# INTERSECTIONZOO: ECO-DRIVING FOR BENCHMARKING MULTI-AGENT CONTEXTUAL REINFORCEMENT LEARNING

**Vindula Jayawardana**[1][*] **Baptiste Freydt**[2][*] **Ao Qu**[1]**, Cameron Hickert**[1]**,**
**Zhongxia Yan**[1]**, Cathy Wu**[1]
[1]MIT, `{vindula, qua, chickert, zxyan, cathywu}@mit.edu`
[2]ETH Zurich, `bfreydt@student.ethz.ch`

## ABSTRACT

Despite the popularity of multi-agent reinforcement learning (RL) in simulated and two-player applications, its success in messy real-world applications has been limited. A key challenge lies in its generalizability across problem variations, a common necessity for many real-world problems. Contextual reinforcement learning (CRL) formalizes learning policies that generalize across problem variations. However, the lack of standardized benchmarks for multi-agent CRL has hindered progress in the field. Such benchmarks are desired to be based on real-world applications to naturally capture the many open challenges of real-world problems that affect generalization. To bridge this gap, we propose *IntersectionZoo*, a comprehensive benchmark suite for multi-agent CRL through the real-world application of cooperative eco-driving in urban road networks. The task of cooperative eco-driving is to control a fleet of vehicles to reduce fleet-level vehicular emissions. By grounding IntersectionZoo in a real-world application, we naturally capture real-world problem characteristics, such as partial observability and multiple competing objectives. IntersectionZoo is built on data-informed simulations of 16,334 signalized intersections derived from 10 major US cities, modeled in an open-source industry-grade microscopic traffic simulator. By modeling factors affecting vehicular exhaust emissions (e.g., temperature, road conditions, travel demand), IntersectionZoo provides one million data-driven traffic scenarios. Using these traffic scenarios, we benchmark popular multi-agent RL and human-like driving algorithms and demonstrate that the popular multi-agent RL algorithms struggle to generalize in CRL settings. Code and documentation are available at https://github.com/mit-wu-lab/IntersectionZoo.

## 1 INTRODUCTION

Having demonstrated impressive performance in simulated multi-agent applications such as Starcraft (Samvelyan et al., 2019), RL holds potential for various multi-agent real-world applications including autonomous driving (Kiran et al., 2021), robotic warehousing (Bahrpeyma & Reichelt, 2022), and traffic control (Wu et al., 2021). However, compared to simulated applications, the success of RL in real-world applications has been rather limited (Dulac-Arnold et al., 2021). A key challenge lies in making RL algorithms generalize across problem variations, such as when weather conditions change in autonomous driving. Problem variations are common in real-world applications but may not be designed to be explicitly assessed in simulated applications (Kirk et al., 2021).

Moreover, there are many open challenges that affect the generalization of RL algorithms in real-world multi-agent problems, such as the effect of complex multi-agent dynamics with aleatory uncertainty and partial observability of states and problem variations, optimizing multiple objectives over long horizons and strict physical constraints. This makes the generalization in multi-agent RL a class of problems rather than one problem as it encapsulates many open challenges (Kirk et al., 2021).

---

[*]Equal contribution.

From an algorithmic point of view, CRL formalizes learning policies that generalize across problem variations, aiming to learn robust, transferable, and adaptable policies (Benjamins et al., 2022). CRL aims to find a policy that can solve a set of (not too different) Markov Decision Processes (MDPs), each posing a problem variation and defined by a variation context vector drawn from a context distribution. This MDP set is referred to as a Contextual MDP (CMDP) (Hallak et al., 2015).

Thus, benchmarking CRL algorithms with real-world applications and the corresponding context distributions could move the needle on the successful use of RL for real-world applications by identifying the many open challenges that affect generalization and measuring progress. However, we observe such efforts are hindered by the lack of standardized multi-agent CRL benchmarks, especially those that are based on real-world applications and that model realistic context distributions. Simulated benchmarks often address only a subset of real-world challenges and lack realistic complexity. Benchmarks based on real-world applications but not actual context distributions still remain partially simulated, adopting a subset of the same limitations. While some multi-agent RL benchmarks can be improvised for CRL benchmarking, this only superficially addresses the underlying problem (Cobbe et al., 2020), lacks desired properties of a CRL benchmark (Kirk et al., 2021) and may lead to conflicting evaluation protocols (Whiteson et al., 2011; Jayawardana et al., 2022). Benchmarks that neglect these properties risk a ceiling effect and fail to incentivize further algorithmic improvements (Ellis et al., 2024; Yu et al., 2022; Hu et al., 2021).

To fill this gap, we present *IntersectionZoo*, a comprehensive multi-agent CRL benchmark suite based on the real-world application of cooperative eco-driving at signalized intersections (Figure 1). Cooperative eco-driving encodes several open challenges that affect generalization in multi-agent CRL, making it well suited for the purpose. IntersectionZoo is built on data-informed context distributions defined by 16,334 signalized intersections in 10 major cities across the United States. By identifying factors affecting vehicular emission (e.g., temperature), IntersectionZoo models nearly one million traffic MDPs on these intersections. Designed with input from domain experts, these MDPs are structured as CMDPs that interface with standard multi-agent interfaces in a highly configurable, fast-to-simulate open-source framework.

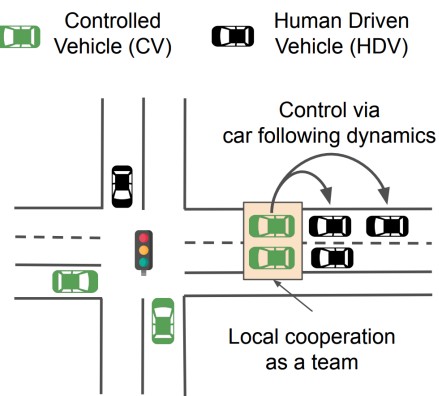

Figure 1: Cooperative eco-driving at signalized intersections where the controlled vehicles (CVs) are operated by an RL policy (or policies) to minimize the fleet-wise emissions that include both CVs and human-driven vehicles (HDVs). CVs implicitly control HDVs through car-following dynamics and form locally cooperative teams for better system control.

IntersectionZoo differs from existing CRL benchmarks on multiple fronts. Contrary to standard autonomous driving benchmarks centered on ego-vehicle control (Li et al., 2022), eco-driving tackles a mixed-motive multi-agent control problem. It is designed based on a well-studied real-world application with known factors that affect generalization (Mintsis et al., 2020; Chen et al., 2022). IntersectionZoo is built on ten CMDPs with data-driven context distributions and also provides the capability to procedurally generate CMDPs. The context distributions of CMDPs are comprehensive, covering variations in states, observations, rewards (with multiple objective terms), and transition dynamics. Last, IntersectionZoo by design supports both independent and identically distributed (IID) and out-of-distribution (OOD) evaluation protocols. We also interface IntersectionZoo with RLLib (Liang et al., 2018) for ease of benchmarking.

**Broader Societal Impact**: Limited familiarity among RL researchers with real-world physical systems has inhibited the development of RL algorithms addressing challenges in these domains. IntersectionZoo bridges this gap by decoupling the modeling complexity of real-world tasks from the experimental process, allowing researchers to focus on algorithmic advancements. This approach, in turn, holds the potential to improve eco-driving, which is known for its impact on climate change mitigation goals (Barkenbus, 2010), with the automotive industry actively pursuing robust eco-driving controllers. This aligns with the growing interest in application-driven research in machine learning, promising a mutually beneficial outcome for all communities involved (Rolnick et al., 2024).

**Remark 1**: With CRL, we focus on intra-task generalization, which means we train and test on environments stemming from the same task (e.g., eco-driving). A related parallel direction is inter-task generalization, training generally capable agents. In this work, we do not focus on that.

## 2 PRELIMINARIES

### 2.1 CONTEXTUAL REINFORCEMENT LEARNING

CRL formalizes the notion of solving a collection of tasks, each stemming from the same problem (e.g., eco-driving). A collection of tasks is formalized using CMDPs (Hallak et al., 2015). We utilize the formalism from Ghosh et al. (2021). Accordingly, a CMDP $\mathcal{M}$ is an MDP with a state space of the form $s = (s', c)$ where $s'$ is the state and $c$ defines the context for the state $s'$. The context $c$ is fixed within an episode (e.g., a fixed random seed). Given an initial context distribution $\rho(c)$, the initial state distribution of a CMDP is defined by $\rho(s) = \rho(c)\rho(s'|c)$. Then, given a context $c$, the CMDP $\mathcal{M}$ is restricted to an MDP $\mathcal{M}_c$ and is called a context-MDP. In other words, the CMDP manifests as a collection of context-MDPs[1]. Note that context $c$ is not always visible to the agents (e.g., a random seed). Then, a CMDP becomes a Contextual Partially Observable MDP (CPOMDP) (Kirk et al., 2021).

We seek to find a policy $\pi^*$ that maximizes the overall expected return across all context-MDPs where $\mathcal{R}(\pi, \mathcal{M}_c)$ is the expected return of policy $\pi$ on context-MDP $\mathcal{M}_c$,

$$\pi^* = \max_{\pi}\Big[\mathbb{E}_{c \sim \rho(c)}[\mathcal{R}\left(\pi, \mathcal{M}_c\right)]\Big]. \tag{1}$$

In multi-agent CRL, each context-MDP $\mathcal{M}_c$ manifests as a multi-agent control problem. Further, these multi-agent control problems are often partially observable. This is also the case with cooperative eco-driving. Then, we leverage a Decentralized Partial Observable MDP (Dec-POMDP) formulation (Bernstein et al., 2002) following previous work (Yan et al., 2022) to define each context-MDP.

### 2.2 COOPERATIVE MULIT-AGENT ECO-DRIVING

Optimizing eco-driving across a full-traffic network is ideal but is impractical in large cities like Los Angeles, with nearly 5000 signalized intersections, and remains an open optimization challenge (Qadri et al., 2020). A common approach is to decompose the network into individual intersections for separate optimizations (Yang et al., 2016; Jayawardana et al., 2024; Yang et al., 2020) while regulating intersection throughput to prevent traffic spill-back due to possible increased throughput. Therefore, it is often assumed that vehicle flow is not at saturation, allowing for reasonable throughput improvements. We adopt the same modeling assumption.

The default objective of cooperative eco-driving at individual signalized intersections is to minimize the total exhaust emissions of a fleet of vehicles (both CVs and HDVs) while having a minimal impact on individual travel time. At a given time $t$, the number of CVs is $k_{CV}^t$, and HDVs is $k_{HDV}^t$ such that $k_{CV}^t + k_{HDV}^t = n^t$ where $n^t$ is the total number of vehicles in the fleet. Then, we control the longitudinal accelerations of all CVs decentrally using a learned policy to optimize,

$$\min J = \sum_{i=1}^{n} \sum_{t=0}^{T_i} E\left(a_i(t), v_i(t)\right) + \lambda T_i. \tag{2}$$

Here, $T_i$ denotes the travel time of vehicle $i$ and time $t$ is a discretized time with increments of $\delta$ (usually 0.5 seconds). The vehicular exhaust emission function is denoted by $E(\cdot)$, which takes speed $v_i(t)$ and acceleration $a_i(t)$ of vehicle $i$ at time $t$ and outputs a vehicular emission amount (usually the amount of carbon dioxide). $\lambda$ is the trade-off hyperparameter. We seek to optimize $J$ subject to hard constraints of ensuring vehicle safety, connectivity via vehicle-to-vehicle and vehicle-to-traffic signal communication, and soft constraints of vehicle kinematics, control realism, traffic safety at the fleet level (e.g., minimum time to collision across all vehicles), and passenger comfort.

---

[1]Here, by MDP we generally refer to any form of MDPs such as Partially Observable MDPs, etc.

**What affects generalization in cooperative eco-driving:** Eco-driving is a complex multi-agent control problem involving both CVs and uncontrolled HDVs where HDVs can introduce uncertainty in CV planning. Adding to the complexity is the required CV coordination. Based on the environment, there can be hundreds of vehicles at an intersection and each vehicle affects the flow of traffic. Further, generalization is naturally required across many factors — such as intersection topology, atmospheric conditions, eco-driving adoption level, and traffic flow rates — defining high-dimensional context distributions. Each CV control relies on a partially observed state of its surroundings and a partially observed context vector. Optimization objectives are multi-fold, with many competing objectives optimized over long horizons. Hard constraints such as vehicle safety have to be satisfied. These challenges individually and collectively shape the generalization capacity of CRL algorithms.

## 3 RELATED WORK

In Table 1, we present our assessment of several known single-agent, multi-agent, and CRL benchmarks, focusing on eleven key properties desired for CRL benchmarking (Kirk et al., 2021; Cobbe et al., 2020; Benjamins et al., 2022). We aim to identify whether they satisfy the given properties or whether repurposing or improvising the current benchmark could be used to satisfy the properties.

*Realistic task* column assesses if a benchmark is based on real-world tasks. Many benchmarks focus on video games (Cobbe et al., 2020; Machado et al., 2018), strategy games (Wang et al., 2021b), grid worlds (Chevalier-Boisvert et al., 2023), or simple control tasks (Benjamins et al., 2022), which lack real-world complexity. This limitation can lead to exploitable structures (Mohan et al., 2024) or hinder algorithmic advancements (Ellis et al., 2024; Yu et al., 2022; Hu et al., 2021). For instance, in the SMAC benchmark, a policy based only on the timestep can achieve notable win rates (Ellis et al., 2024). Some work involves more realistic robotics tasks (James et al., 2020; Yu et al., 2020), but they often use tightly constrained context features like limited friction levels.

*Data-driven context distribution* checks if benchmarks provide real-world context distributions, while *native CRL support* assesses if a benchmark is primarily designed to support CRL. Except for MetaDrive (Li et al., 2022) and SMACv2 (Ellis et al., 2023), most native CRL benchmarks focus on single-agent CRL. MetaDrive is a close second to IntersectionZoo but lacks data-driven context distributions for CMDPs, limiting its ability to capture real-world complexity. Moreover, the *initializable context distributions* column checks if a benchmark facilitates initializing arbitrary user-defined context distributions. IntersectionZoo facilitates using both additional real-world context distributions (e.g., intersections of other cities) as well as procedurally generating contexts based on user-specified context feature distributions.

Typically, there are two types of features defining the context distribution and thereby describing the context of a context-MDP: *observed context features* and *unobserved context features*. The key difference is whether these features are explicitly visible to the agent. Observed features are directly visible factors of variations, such as lane length in eco-driving. Unobserved features are mostly used for random variations such as those that arise from procedural content generation (PCG) (Cobbe et al., 2020) and are not explicitly visible to the agent (e.g., HDV aggressiveness). Both observed and unobserved features are desired for CRL benchmarking (Kirk et al., 2021). This enables systematic targeted evaluations of generalization such as *systematicity* (generalization using systematic recombination of known knowledge) and *productivity* (generalization beyond seen training data) (Hupkes et al., 2020) with non-trivial tasks.

*Varying $\mathcal{S}, \mathcal{O}, \mathcal{T}, \mathcal{R}$* refers to variations in states ($\mathcal{S}$), observations ($\mathcal{O}$), transitions ($\mathcal{T}$), and rewards ($\mathcal{R}$). In IntersectionZoo, the variations in environments stem from states (e.g., single vs. multiple lane driving), observations (e.g., diverse vehicle sensor capabilities), rewards (e.g., the trade-off between travel time and emission reduction), and dynamics (e.g., vehicle behavior changes due to varying traffic signal timings). Multiple forms of variations enable diversity and provide more avenues for targeted evaluations of generalizations (Li et al., 2022). While benchmarks like CARL (Benjamins et al., 2022) and MDP Playground (Rajan et al., 2023) are categorized to consist of all forms of variations, they lack evaluation protocols spanning all these variations or consist of multiple tasks without including all variations in each task.

*Multiple objectives* can bring another form of variation in CMDPs and are common in real-world problems. It thus manifests as another axis of targeted evaluation of generalization. While existing multi-objective RL benchmarks overlook generalization challenges (Felten et al., 2024), addressing

Table 1: Comparison of IntersectionZoo to related benchmarks. Benchmarks are rated based on whether they satisfy (✔), somewhat satisfy (🟡), or do not satisfy (✘) a given desired property. 'Context' is abbreviated as 'ctxt.' for formatting.

| Benchmark | Realistic task | Multi-agent | Observed ctxt. features | Unobserved ctxt. features | Initializable ctxt. dist. | Varying $\mathcal{S}, \mathcal{O}, \mathcal{T}, \mathcal{R}$ | Multi-objective | IID testing | OOD testing | Data-driven ctxt. dist. | Native CRL support |
|---|---|---|---|---|---|---|---|---|---|---|---|
| MDP Playground (Rajan et al., 2023) | ✘ | ✘ | ✔ | ✔ | ✘ | ✔ | ✘ | ✘ | ✘ | ✘ | ✘ |
| bsuite (Osband et al., 2020) | ✘ | ✘ | ✔ | ✘ | 🟡 | ✘ | ✘ | ✘ | ✘ | ✘ | ✘ |
| Revisiting ALE (Machado et al., 2018) | ✘ | ✘ | 🟡 | 🟡 | ✘ | ✘ | ✘ | ✘ | ✘ | ✘ | ✘ |
| CARL (Benjamins et al., 2022) | ✘ | ✘ | ✔ | ✘ | ✔ | ✔ | ✘ | ✔ | ✔ | ✘ | ✔ |
| Procgen (Cobbe et al., 2020) | ✘ | ✘ | ✘ | ✔ | ✘ | 🟡 | ✘ | ✔ | ✘ | ✘ | ✔ |
| Alchemy (Wang et al., 2021b) | ✘ | ✘ | ✘ | ✔ | ✘ | ✔ | ✘ | ✔ | 🟡 | ✘ | ✔ |
| Meta-World (Yu et al., 2020) | 🟡 | ✘ | ✔ | 🟡 | ✔ | ✔ | ✔ | ✔ | 🟡 | ✘ | ✔ |
| RL Bench (James et al., 2020) | 🟡 | ✘ | ✔ | ✔ | ✔ | ✔ | ✘ | ✔ | ✔ | ✘ | ✔ |
| Minigrid (Chevalier-Boisvert et al., 2023) | ✘ | ✘ | ✔ | ✔ | ✘ | 🟡 | ✘ | ✘ | 🟡 | ✘ | ✔ |
| NetHack (Küttler et al., 2020) | ✘ | ✘ | ✘ | ✔ | ✘ | ✔ | 🟡 | ✔ | ✔ | ✘ | ✔ |
| MiniHack (Samvelyan et al., 2021) | ✘ | ✘ | ✔ | ✔ | ✔ | ✔ | 🟡 | ✔ | ✔ | ✘ | ✔ |
| Particle Env (Lowe et al., 2017) | ✘ | ✔ | 🟡 | ✔ | ✘ | ✔ | ✘ | ✔ | 🟡 | ✘ | ✘ |
| SMACv2 (Samvelyan et al., 2019) (Ellis et al., 2023) | ✘ | ✔ | ✔ | ✔ | ✔ | 🟡 | ✘ | ✔ | 🟡 | ✘ | ✔ |
| Google Football (Kurach et al., 2020) | ✔ | ✔ | ✔ | ✔ | ✘ | ✘ | ✘ | ✘ | ✘ | ✘ | ✘ |
| Flow (Wu et al., 2021) | 🟡 | ✔ | ✔ | ✔ | ✔ | ✔ | ✘ | ✔ | 🟡 | ✘ | ✘ |
| Nocturne (Vinitsky et al., 2022) | ✔ | ✔ | ✔ | 🟡 | ✘ | ✔ | 🟡 | ✔ | ✔ | ✘ | ✘ |
| MetaDrive (Li et al., 2022) | ✔ | ✔ | ✔ | ✔ | ✔ | 🟡 | ✘ | ✔ | ✔ | ✘ | ✔ |
| SMARTS (Zhou et al., 2021) | ✔ | ✔ | ✔ | 🟡 | 🟡 | ✔ | 🟡 | ✔ | 🟡 | ✘ | ✘ |
| RESCO (Ault & Sharon, 2021) | ✔ | ✔ | ✔ | ✘ | ✘ | 🟡 | 🟡 | 🟡 | 🟡 | ✔ | ✘ |
| IntersectionZoo (ours) | ✔ | ✔ | ✔ | ✔ | ✔ | ✔ | ✔ | ✔ | ✔ | ✔ | ✔ |

this gap is acknowledged as needed (Kirk et al., 2021). Nocturne (Vinitsky et al., 2022), Flow (Wu et al., 2021) and SMAC (Ellis et al., 2023; Samvelyan et al., 2019) can be improvised for this purpose but lack native support. IntersectionZoo naturally frames a multi-objective optimization problem, prioritizing emission reduction with less impact on vehicle travel time. Further sub-objectives can incorporate passenger comfort, kinematic realism, and traffic level safety (Appendix A.6).

*IID and OOD testing* are commonly used as evaluation protocols of generalization. IID evaluation assumes training and testing context sets are drawn from the same distribution. Benchmarks such as Procgen (Cobbe et al., 2020) or NetHack (Küttler et al., 2020) employ this scheme, where the training seeds can be sampled uniformly at random from the full distribution, and the full distribution can be used for testing. OOD evaluation does not assume the same distribution for training and testing and requires domain generalization (Wang et al., 2022). IntersectionZoo models 10 different CMDPs based on 10 US cities, each with a different intersection distribution. Hence, OOD evaluation can be performed by training on one CMDP and testing on another. Similarly, IID testing can be performed by train/test split within a CMDP.

## 4 INTERSECTIONZOO

Designing a CRL benchmark suite based on a real-world task is challenging. Identifying the factors of variations requires domain knowledge and expert opinion, and data-driven modeling is required

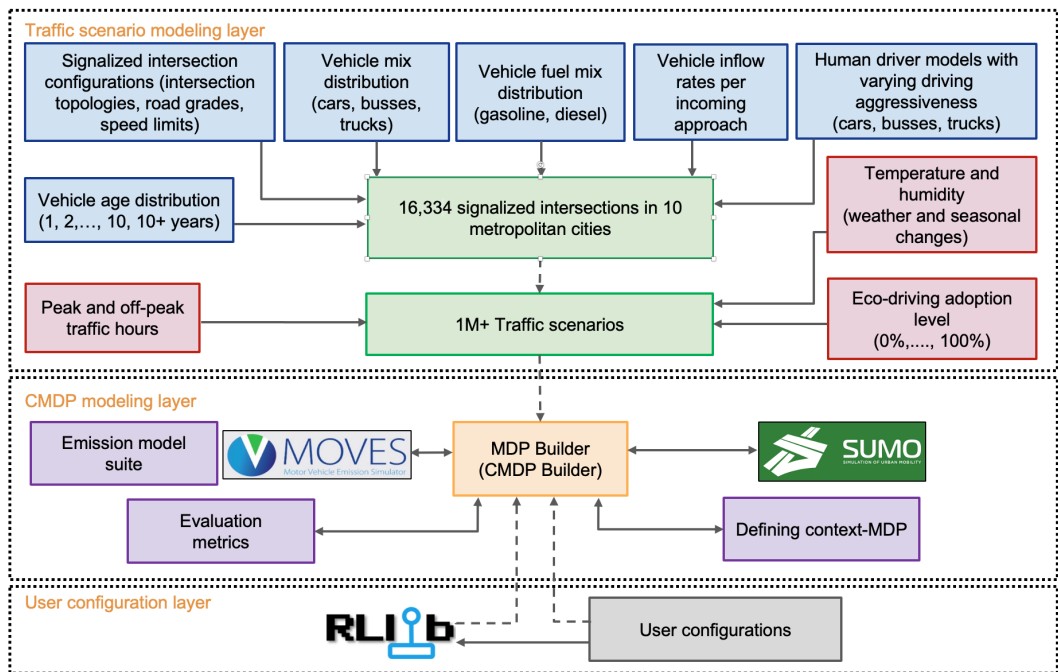

Figure 2: A schematic overview of IntersectionZoo divided into three architectural layers.

to model them realistically without artificially making the problem easy or hard. For example, it is known that eco-driving in short lanes is hard (Jayawardana et al., 2024), and hence, a context distribution that may have superficially many short lanes could make the problem harder than it is. In the following sections, we provide details of how we construct IntersectionZoo with the help of domain experts. We refer the reader to Appendix A.9 for more detials on release notes of IntersectionZoo including license details.

## 4.1 DEFINING INTERSECTIONZOO SCENARIOS FOR CONSTRUCTING CMDPs

In IntersectionZoo eco-driving CMDPs are formulated as a collection of traffic context-MDPs. Each traffic scenario is a basis for a context-MDP. A traffic scenario manifests as a combination of a set of eco-driving factor values that have a known effect on emission benefits at signalized intersections. These factors are related to intersection topology, human driver behavior, vehicle characteristics, traffic flow, and atmospheric conditions.

Concretely, an intersection is first defined by factors such as lane lengths, lane counts, road grades, turn lane configuration, and speed limit of each approach. Then, vehicle type, age, and fuel type distributions are used with appropriate traffic flow rates and HDV behaviors to define a realistic traffic flow. Each intersection scenario is further assigned representative atmospheric temperature and humidity values based on the season. Further scenario variations can be achieved by changing the eco-driving adoption level (0%-100%). We follow this procedure for every intersection, and the resultant traffic scenarios are the basis for the context-MDPs of each city.

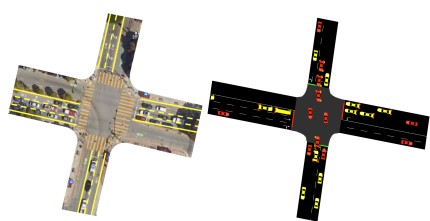

Figure 3: **Left**: The signalized intersection in the intersection of Bosworth Street and Diamond Street in Salt Lake City, Utah. **Right**: The reconstructed intersection in simulation.

Figure 2 illustrates the overall process: the factors we consider and how they are leveraged to build intersections, traffic scenarios, and, subsequently, CMDPs. In the *traffic scenario modeling layer*, we first build data-driven simulation environments of signalized intersections. We use Open Street Maps (OSM) (Haklay & Weber,

2008) data and follow guidelines provided by Qu et al. (2022). Intersection lane lengths, lane counts, turn lane configurations, and speed limits are extracted from OSM. Road grades are taken from US geological surveys (Survey). An example instance of a reconstructed intersection is given in Figure 3. More details are given in Appendix A.1 regarding all ten cities and the feature distributions.

To model the vehicle arrival process, we use the Annual Average Daily Traffic data (AADT) (Huntsinger, 2022) released by the Departments of Transportation of each state/city. To differentiate the inflows between peak and off-peak hours, we use recommended conversion rates (Precisiontraffic, 2014). To model realistic vehicle arrival processes to intersections, we also model one-hop nearby intersections as illustrated in Figure 4.

For traffic signal timing, we exhaustively search through the fixed-time traffic signal timing plans (Thunig et al., 2019) and find the optimal plan. We source vehicle age, fuel type, and vehicle type distributions from the openly available MOVES databases (epa) and data from US National Centers for Environmental Information (for Environmental Information) is used for atmospheric condition modeling with temperature and humidity changes.

To model HDV behavior, we use the Intelligent Driver Model (IDM) (Treiber et al., 2000) and calibrate it using the guidelines from Zhang & Sun (2022) with real-world arterial driving data from CitySim (Zheng et al., 2022) (See Appendix A.2 for details). While IDM controls the longitudinal acceleration of HDVs, lane changes are performed using standard rule-based lane changing model (Erdmann, 2014).

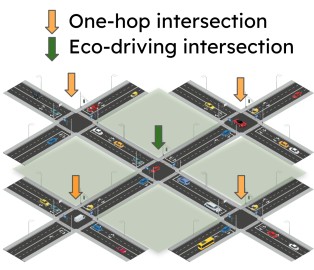

Figure 4: For each intersection, default nearby intersections are added for realistic vehicle arrival processes subjected to nearby traffic signals.

In the *CMDP modeling layer*, we use the modeled traffic scenarios and organize them based on the city to create 10 CMDPs. All traffic scenarios are configured for use in the open-source agent-based traffic simulator SUMO (Lopez et al., 2018). A key requirement for capturing the effect of traffic scenarios on vehicle exhaust emission is a rich emission function. For this purpose, we integrate a suite of comprehensive and fast neural emission models (Sanchez et al., 2022) that replicate the industry-standard Motor Vehicle Emission Simulator (MOVES) (epa). Finally, the *user configuration layer* exposes the CMDPs that can be used by researchers for training CRL algorithms. We provide RLLib-based training by default and also enable flexibility with custom implementations.

**A note on simulation realism**: Autonomous driving benchmarks often prioritize the realistic rendering of traffic scenarios, aiming to train closed-loop policies from sensory inputs like perception inputs to navigate the ego-vehicle. However, in cooperative eco-driving, we use a 2D simulator that generates vectorized representations of vehicle states, isolating the challenge of generalization in CRL over feature extraction. Given that CRL often requires many samples during training, fast simulations are also an essential requirement that we achieve through simplified rendering.

## 4.2 DEFINING CONTEXT-MDPS

Here, we provide an overview of the context-MDP definition we use in IntersectionZoo and refer the reader to Appendix A.4 for more specific details. In cooperative eco-driving, each traffic scenario manifests as a partially observable multi-agent control problem. Then, we leverage Decentralized Partial Observable MDP (Dec-POMDP) formulation (Bernstein et al., 2002) following previous work (Yan et al., 2022) to define each context-MDP. For each CV, state, action, and reward are defined as follows.

**State Space**: The design of the observed state of a vehicle is mainly based on the capabilities of existing sensor technologies. The observed state of a CV includes its own status, status of neighboring vehicles (leader and follower on all immediate nearby lanes), and status of the immediate traffic signal timing. Further, we provide a selected set of observed features based on the feasibility of obtaining them in the real world by the CV as observed context of the underlying environment.

**Action Space**: Longitudinal acceleration of each CV. Standard rule-based controller is used (Erdmann, 2014) for lane changing, focusing IntersectionZoo on the continuous control aspect of eco-driving.

**Multi-objective Reward Function**: The reward $r_i^t$ for each CV $i$ at time $t$ is defined in Equation 10 inspired by Jayawardana et al. (2024). Here, $n_t$ is the vehicle fleet size, $v_t^i$ is the velocity, and $e_t^i$ is the $CO_2$ emissions of vehicle $i$ at time $t$. Hyperparameters include, $\eta = [0, 1]$, $\alpha$, $\beta$, and $\tau$. The indicator function $\mathbb{1}_{v_t^i < \tau}$ indicates whether the vehicle is stopped, while the term $e_t^i$ encourages low emissions. The velocity term captures the effect on travel time. Users can configure the parameter $\eta$ to either get a fleet-based reward, agent-based reward, or a combination of both. All such formulations are acceptable.

$$r_t^i = \eta \frac{1}{n_t} \sum_{j=1}^{n_t} (v_t^j + \alpha \mathbb{1}_{v_t^j < \tau} + \beta e_t^j) + (1 - \eta)(v_t^i + \alpha \mathbb{1}_{v_t^i < \tau} + \beta e_t^i). \tag{3}$$

Appendix A.6 details additional sub-objectives in IntersectionZoo to enhance fleet-level traffic safety, passenger comfort, and kinematic realism. IntersectionZoo ensures vehicle safety and actuator limits via pre-defined rule-based checks when used with the default objective.

# 5 EVALUATIONS IN INTERSECTIONZOO

## 5.1 EVALUATION PROTOCOLS

By default, IntersectionZoo provide interfaces for train/test split evaluations to measure generalization, which is often used with zero-shot policy transfer (Harrison et al., 2019; Higgins et al., 2017; Kirk et al., 2021). This means we train policies on one subset of context MDPs and test on another subset of context MDPs. This includes both IID and OOD evaluation protocols. Hence, OOD evaluation can be performed by training in one city (train CMDP) and testing in another city (test CMDP). Similarly, IID testing can be performed by train/test split of context-MDPs within a given city.

## 5.2 BASELINES

**IDM**: IntersectionZoo provides calibrated Intelligent Driver Models using real-world human driving data (Appendix A.2) as a human driving baseline for benchmarking the performance of an eco-driving CRL policy. The goal here is to capture how much emission reduction can be obtained from driving differently than humans usually do (Jayawardana & Wu, 2022; Jayawardana et al., 2024).

**GLOSA**: IntersectionZoo provides an implementation of commercially used state-of-the-art GLOSA eco-driving controller (Katsaros et al., 2011) for performance comparison. This is a heuristical approach for eco-driving that has been implemented by popular car companies such as Audi (aud). The core idea of GLOSA is simple – avoid idling by gliding to the intersection while the signal is red. The basic GLOSA controllers do not account for the impact of nearby vehicles when deciding the gliding deceleration for each ego agent, which may make it sub-optimal when vehicle inflow is high.

## 5.3 EVALUATION METRICS

**Individual metrics**: We assess the performance of a policy using the average vehicular exhaust emission per vehicle and the intersection throughput of each intersection. Refer to Appendix A.7 for more details on the definitions of these metrics.

**Composite metric**: We introduce *effective emission benefits* as a composite metric that aggregates the individual metrics across multiple intersection approaches for method ranking. A crucial criterion for evaluating the performance of an eco-driving policy is to ensure that it does not reduce intersection throughput, as detailed in Section 2.2. Effective emission benefits quantify the emission improvements across intersection approaches while considering only approaches where both emissions and throughput show positive gains. For any approach where throughput or emissions deteriorate compared to the IDM controller, the policy reverts to human-like driving at that approach using IDM, resulting in zero benefits for both metrics. Formally, we define the effective emission per vehicle $e_\pi^a$ for an approach $a \in A$ under policy $\pi$ as,

$$e_\pi^a = \begin{cases} e_\pi^a & \text{if } N_a \geq 0 \ \& \ E_a \geq 0 \\ e_{idm}^a & \text{otherwise} \end{cases} \tag{4}$$

where $E_a$ and $N_a$ denote the two individual metrics: emission benefit percentage and throughput benefit percentage as described in Appendix A.7, respectively. Then, the metric *effective emission benefit* is defined as a percentage improvement,

$$E(\%) = 100 \cdot (\sum_{a \in A} e_{\text{idm}}^a - \sum_{a \in A} e_\pi^a) / \sum_{a \in A} e_{\text{idm}}^a \qquad (5)$$

## 6 INTERSECTIONZOO BENCHMARKING

In this section, we benchmark popular RL algorithms used in multi-agent control with three objectives. First, we aim to show current MARL methods do not appear to easily solve the CRL problem in IntersectionZoo CMDPs and hence indicate that IntersectionZoo provides reasonable room for future CRL algorithmic improvements. Second, while each CMDP is not fully solved, we show that some context-MDPs in both Atlanta and SLC CMDPs are solved by these algorithms, indicating that IntersectionZoo design is reasonably tuned and MARL methods can learn and perform up to a certain extent. Third, we aim to show example use cases of IntersectionZoo and demonstrate how it can be used to perform targeted evaluations of generalizations - an aspect that has not been focused on by previous benchmarks.

Considering their success in many cooperative multi-agent problems (Yu et al., 2022), we employ PPO (Schulman et al., 2017), DDPG (Silver et al., 2014), multi-agent PPO (MAPPO) and DDPG (MADDPG) with a centralized critic (Yu et al., 2022; Lowe et al., 2017), and graph convolution networks (GCRL) for explicit cooperation modeling in multi-agent RL (Jiang et al., 2020). GLOSA baseline controller is also used for comparison. All performance benefits are given relative to the IDM controller. According to our benchmarking goals, we mainly assess their IID generalization capacity but briefly test two more forms of targeted evaluations of generalizations: systematicity and productivity, to demonstrate the use of IntersectionZoo. Due to the space limitation, the systematicity and productivity analysis is given in Appendix B.4. We illustrate the emission benefits at the level of intersection incoming approaches instead of the intersection for a better understanding and visualization of the benefit distributions.

| Method | Atlanta effective emission benefits ($\uparrow$) | SLC effective emission benefits ($\uparrow$) |
|--------|---------------------------------------------------|----------------------------------------------|
| GLOSA | 2.30% | 0.95% |
| PPO | 0.47% | 0.05% |
| DDPG | **2.84%** | **2.95%** |
| MAPPO | 1.11% | 0.61% |
| GCRL | 0.0% | 0.0% |
| MADDPG | **4.47%** | 0.0% |

Table 2: Effective emission benefits of different methods across Atlanta and Salt Lake City(SLC) CMDPs. All percentages are given relative to human-like driving baselines given by IDM policy. Bolded values indicate performances better than the GLOSA baseline.

We assess how well the MARL algorithms can generalize when the training and testing CMDPs are the same (in-distribution generalization). For this, we leverage Atlanta CMDP (621 intersections) and Salt Lake City (SLC) CMDP (282 intersections) under summer temperature and humidity, with 1/3rd of vehicles being CVs. Following the IID evaluation protocol, we train and evaluate on the same CMDP. The results are given in Table 2 for effective emission benefits. As can be seen, most of the algorithms achieve positive emission benefits overall[2]. DDPG and MADDPG in Atlanta and DDPG in SLC perform better than the GLOSA controller, indicating that MARL can, in fact, learn better policies than simple rule-based methods. These results serve to validate one of our objectives of benchmarking - the IntersectionZoo context-MDP definitions (e.g., reward and observation functions) are reasonable, and MARL methods can learn to improve the emissions.

However, while the effective emission benefits are positive for most MARL methods, some context-MDPs are not solved in each CMDP. Emission benefit histograms given in Figure 5 for Atlanta further

---

[2]We typically do not expect significantly high effective emission benefits percentages based on eco-driving literature. However, when done at a large scale, these emission reductions still result in large $CO_2$ reductions.

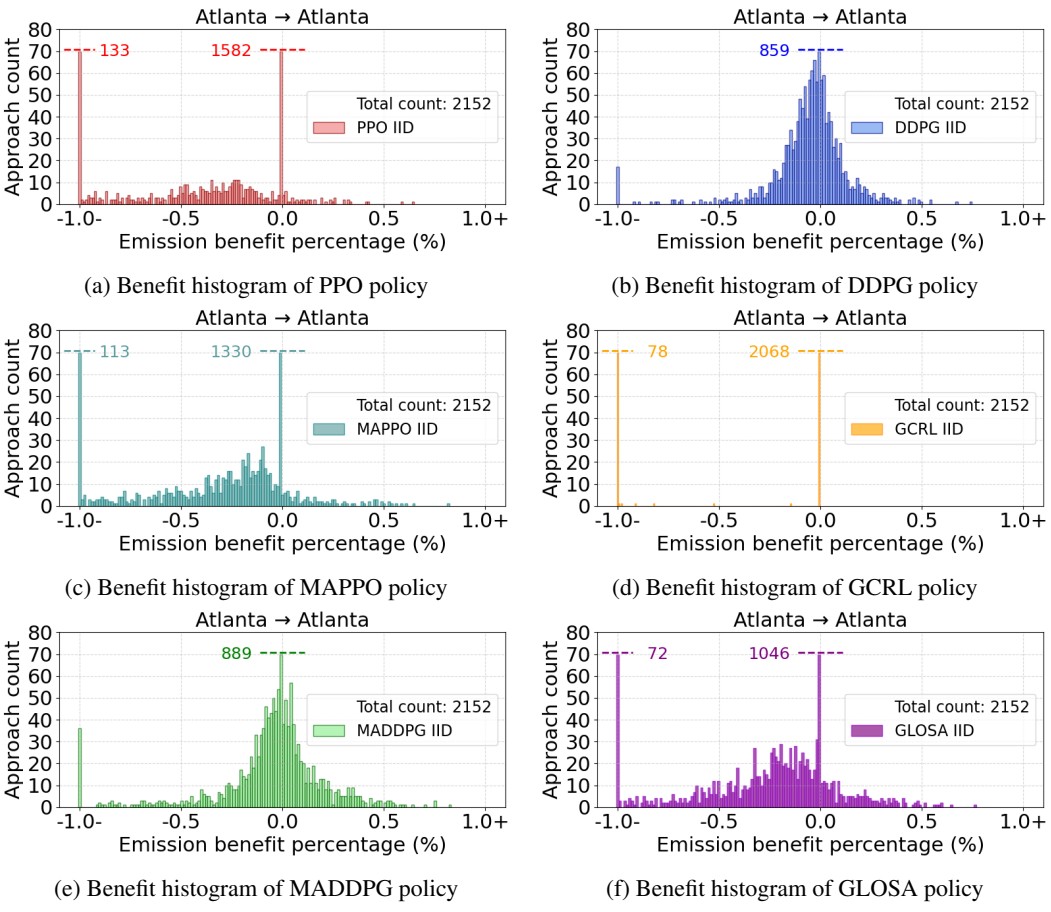

Figure 5: Emission benefit histograms of Atlanta under different RL algorithms with in-distribution testing. Percentages are relative to the IDM baseline. Large y-axis counts are truncated for clarity. The total approach count refers to the total number of intersection approaches. The spikes at 0% are in part due to the zeroing of emissions benefits for any scenarios where throughput is reduced.

illustrate the benefit distributions. The SLC histogram is given in Appendix Section B.3. This serves as evidence of the fact that current MARL methods do not appear to easily solve the CRL problem in IntersectionZoo CMDPs and hence indicate that IntersectionZoo provides reasonable room for future CRL algorithmic improvements.

We also show example use cases of IntersectionZoo when used for targeted evaluations of generalizations - an aspect neglected by most existing benchmarks. We test systematicity and productivity of generalization (Kirk et al., 2021; Hupkes et al., 2020). Due to the space limitations, we refer the reader to Appendix Section B.4 for more details of this analysis and results.

## 7 CONCLUSION

In this work, we propose IntersectionZoo, a comprehensive multi-agent CRL benchmark suite based on the real-world application of cooperative eco-driving. Using IntersectionZoo, we benchmark multi-agent RL, rule-baed, and human-like driving algorithms and demonstrate that the popular multi-agent RL algorithms struggle to effectively generalize in CRL settings. A current limitation of the benchmark is it can be primarily used to benchmark only continuous control algorithms. However, IntersectionZoo provides discrete lane-changing control for interested users. Further, despite our best efforts to create realistic traffic scenarios, the provided scenarios may have variations from their real-world counterparts due to inevitable data errors and missing data. Overall, IntersectionZoo aims to advance generalization in multi-agent RL research by providing a rich benchmark suite that naturally captures many of the real-world problem characteristics that affect generalization.

## 8 ACKNOWLEDGMENTS

The authors thank Blaine Lenoard, Mark Taylor, Michael Sheffield, Christopher Siavrakas, and Kelly Njord at Utah Department of Transportation for their constructive feedback on eco-driving scenarios. The authors acknowledge the MIT SuperCloud and Lincoln Laboratory Supercomputing Center for providing computational resources to conduct experiments reported in this paper.

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

# Appendices

# A APPENDIX A - ADDITIONAL DETAILS

## A.1 DETAILS OF INTERSECTIONZOO CMDP CONTEXT DISTRIBUTIONS

IntersectionZoo provides 10 CMDPs based on 10 major metropolitan cities across the United States. In Table 3, we list the ten cities we consider and the number of intersections considered for building these CMDPs in each city. In Figure 8, we present the spatial distributions of the intersections considered in building the IntersectionZoo across the ten cities.

| City | Intersection Count |
|---|---|
| Atlanta | 621 |
| Boston | 917 |
| Chicago | 2864 |
| Dallas Fort Worth | 1570 |
| Houston | 1014 |
| Los Angeles | 4276 |
| Salt Lake City | 282 |
| New York Manhattan | 2586 |
| San Francisco | 1209 |
| Seattle | 995 |
| Total | 16,334 |

Table 3: Intersection count distribution within the ten cities.

In Figure 6, we provide a comparison of feature distributions of the incoming approaches of intersections of the ten cities. The y-axis is the percentage, while each x-axis is the respective feature range. Lane length is measured in meters, speed limit is in meters per second, vehicle inflows are vehicles per hour, and road grade is in percent grade. Phase count denotes how many different traffic signal phases are applicable for a given incoming approach. The signal time ratio denotes the ratio between approach-related phase times and total cycle time.

Based on our traffic scenario modeling efforts, IntersectionZoo provides more than one million traffic scenarios. As a high-level breakdown, for all 16,334 intersections, we enable modeling four seasons as changes in temperature and humidity, four main eco-driving adoption levels (10%, 20%, 50%, 75%, and 100%), two traffic hours (peak and off-peak), two-vehicle engine technologies (electric vs. internal combustion engine) summing up to more than 1 million traffic scenarios.

In Figure 7, we illustrate some representative intersection topologies reconstructed in IntersectionZoo.

## A.2 MODELING HUMAN-DRIVEN VEHICLES

For the foreseeable future, human-driven vehicles will remain prevalent. To model human drivers in our simulations, we use the Intelligent Driver Model (IDM) (Treiber et al., 2000). IDM is a widely accepted car-following model that can produce realistic traffic waves. IDM calculates a vehicle's acceleration using Equation 6, with desired velocity $v_0$, space headway $s_0$, time headway $T$, maximum acceleration $\alpha$, and comfortable braking deceleration $\beta$. The velocity difference with the leading vehicle is denoted as $\Delta v(t)$, and $\delta$ is a constant.

$$a(t) = \alpha \left[ 1 - \left( \frac{v(t)}{v_0} \right)^\delta - \left( \frac{s^*(v(t), \Delta v(t))}{s(t)} \right)^2 \right] \tag{6}$$

$$s^*(v(t), \Delta v(t)) = s_0 + \max \left( 0, v(t)T + \frac{v(t)\Delta v(t)}{2\sqrt{\alpha\beta}} \right) \tag{7}$$

For simulation accuracy, precise calibration of parameters $v_0$, $s_0$, $T$, $\alpha$, and $\beta$ is crucial. Different regions may exhibit varying driving behaviors, such as American drivers versus British drivers. Thus, calibrating the IDM parameters is critical for accurate human driver modeling.

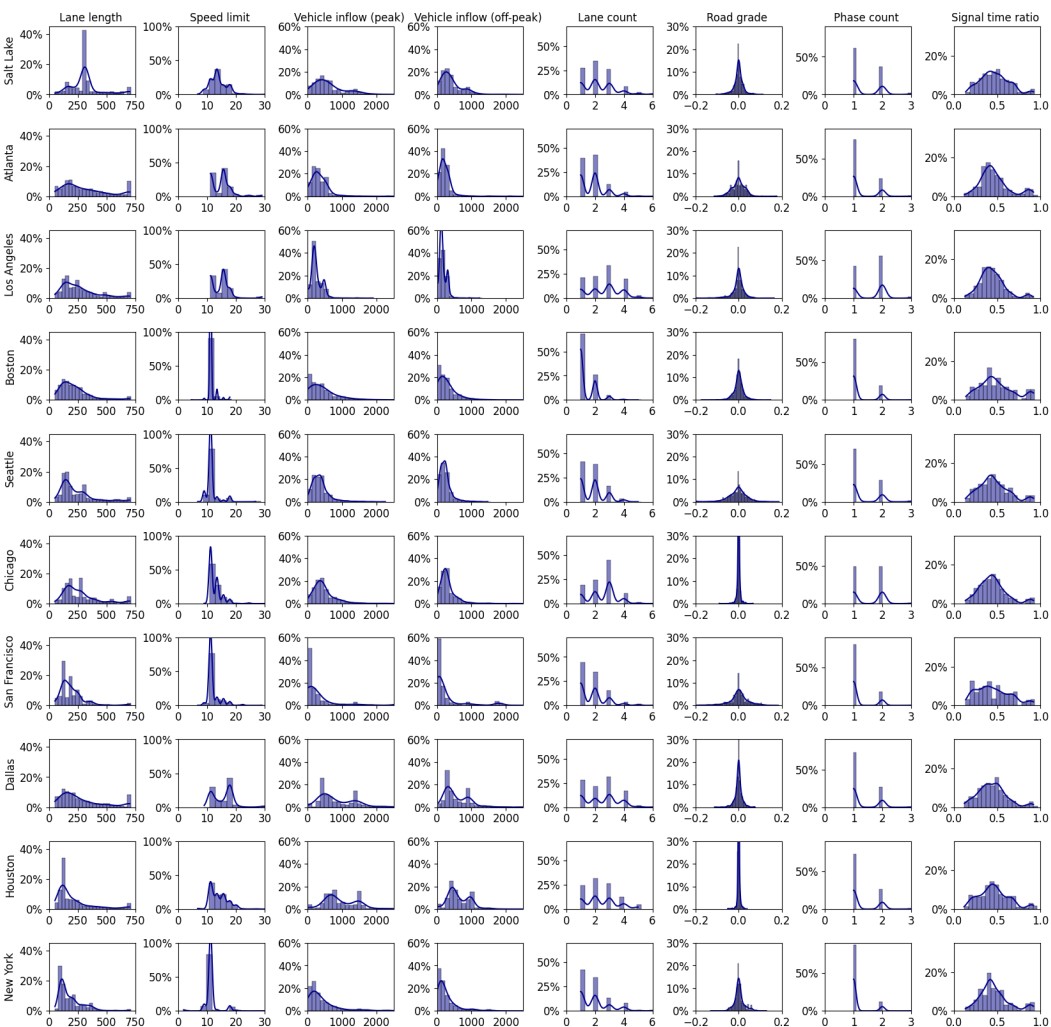

Figure 6: Comparison of context feature distributions of the incoming approaches of the ten cities. The y-axis is the percentage, while each x-axis is the respective feature range.

Our IDM model calibration goals are threefold. We aim to align it with real-world human driving behavior at US signalized intersections, tailoring the five IDM parameters for human-like trajectories. We also need separate IDM models for distinct vehicle types (e.g., cars, buses, trucks) due to their influence on driving style. Lastly, we aim to establish a range of IDM models reflecting diverse driver behaviors and aggressiveness as observed in real-world driving.

For this purpose, we employ Bayesian inference and Gaussian process-based approach proposed by Zhang & Sun (2022) and leverage real-world arterial driving data from CitySim (Zheng et al., 2022). For $\forall (t, d) \in \{(t, d)\}_{t=t_0, d=1}^{t_0+(T-1)\Delta t, D}$ where $d$ represents the index for each driver $d \in \{1, \cdots, D\}$ and $t$ represents the timestamp, we have,

$$\ln(\theta) \sim \mathcal{N}(\mu_0, \Sigma_0) \in \mathbb{R}^5 \tag{8a}$$

$$\ln(\sigma_\epsilon) \sim \mathcal{N}(\mu_\epsilon, \sigma_1) \in \mathbb{R} \tag{8b}$$

$$v_d^{(t+\Delta t)} \mid i_d^{(t)}, \theta \overset{\text{i.i.d.}}{\sim} \mathcal{N}\left(\mathcal{F}_{\text{IDM}}\left(i_d^{(t)}; \theta\right), (\sigma_\epsilon \Delta t)^2\right) \in \mathbb{R} \tag{8c}$$

Here, $\theta = [v_0, s_0, T, \alpha, \beta] \in \mathbb{R}^5$ is the IDM model parameters where $\mathcal{N}(\mu, \sigma)$ represents a Gaussian distribution with a mean of $\mu$ and standard deviation of $\sigma$. Independent and identically distributed is

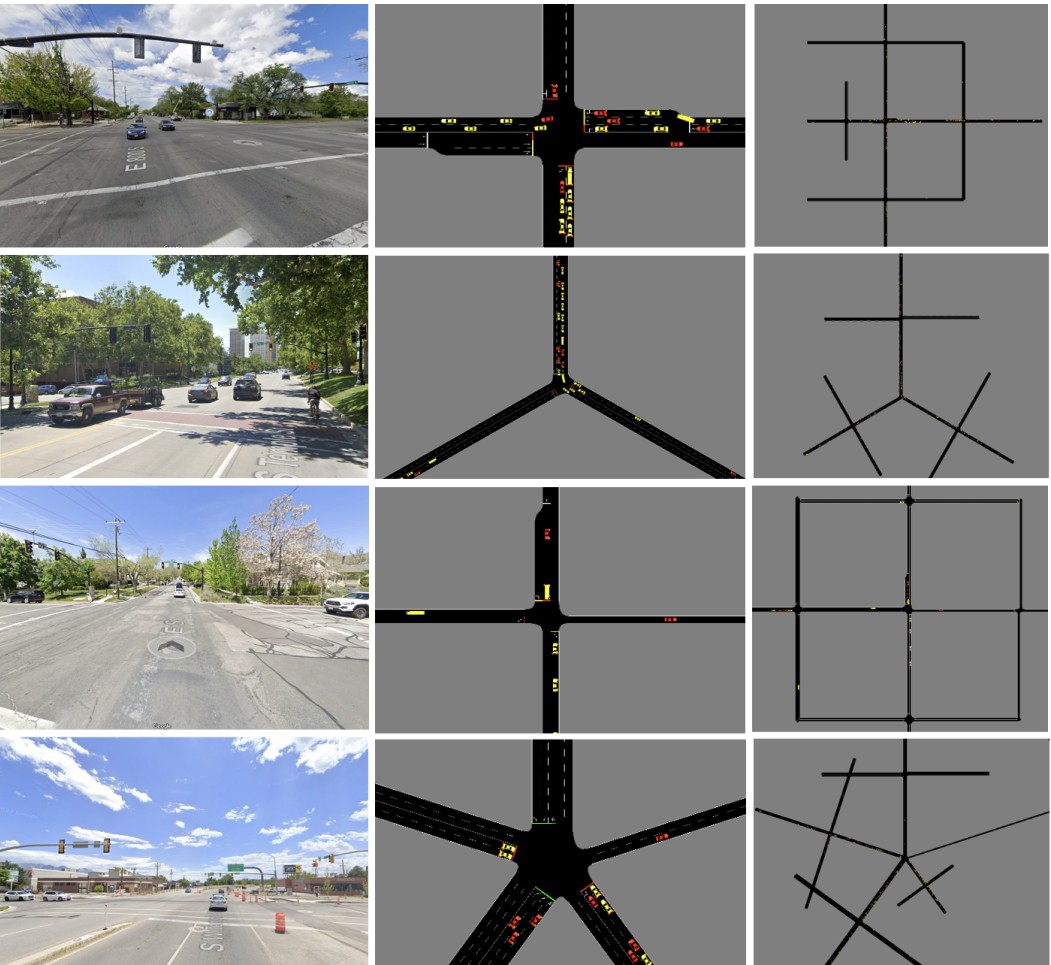

Figure 7: **Left**: a snapshot of the real-world intersection. **Middle**: the reconstructed intersection topology in SUMO. **Right**: the full network around the ego-intersection with one-hop nearby intersections added for realistic vehicle arrival processes subjected to nearby traffic signals. IntersectionZoo covers many of the diverse intersection topologies as briefly illustrated here.

indicated by $i.i.d.$ Furthemore, we denote the inputs at time $t$ as a vector $i_d^{(t)} = [s_d^{(t)}, v_d^{(t)}, \Delta v_d^{(t)}], \forall t \in \{t_0, \cdots, t_0 + (T-1)\Delta t\}$ where $s_d^{(t)}$ is the headway, $v_d^{(t)}$ is the velocity and $\Delta v_d^{(t)}$ is the relative velocity of driver $d$ and its leading vehicle at time $t$. We further define a function $\mathcal{F}_{IDM}(\cdot)$ that updates the ego vehicle's speed at $t + \Delta t$ using Equation 6 and Equation 9 where $\Delta t$ is the step size.

$$v^{(t+\Delta t)} = v^{(t)} + a^{(t)}\Delta t \tag{9}$$

The formulation in Equations 8 generates a population-level joint distribution of IDM parameters. As vehicle type influences driving behavior, we build distinct joint distributions for each vehicle type. This requires separate human-driving trajectory datasets for every vehicle type.

CitySim (Zheng et al., 2022) is a drone-based vehicle trajectory dataset that provides vehicle trajectories along arterial roads, but it lacks explicit vehicle type labels. Therefore, we use the bounding box length of vehicles to distinguish cars from buses and trucks. While this identifies cars, it does not allow us to differentiate between trucks and buses. Thus, we assume a common IDM parameter distribution for trucks and buses. This is reasonable as both are heavy-weight vehicles with similar behavior at signalized intersections. After modeling these joint distributions, we utilize them to sample human drivers for the micro-simulations.

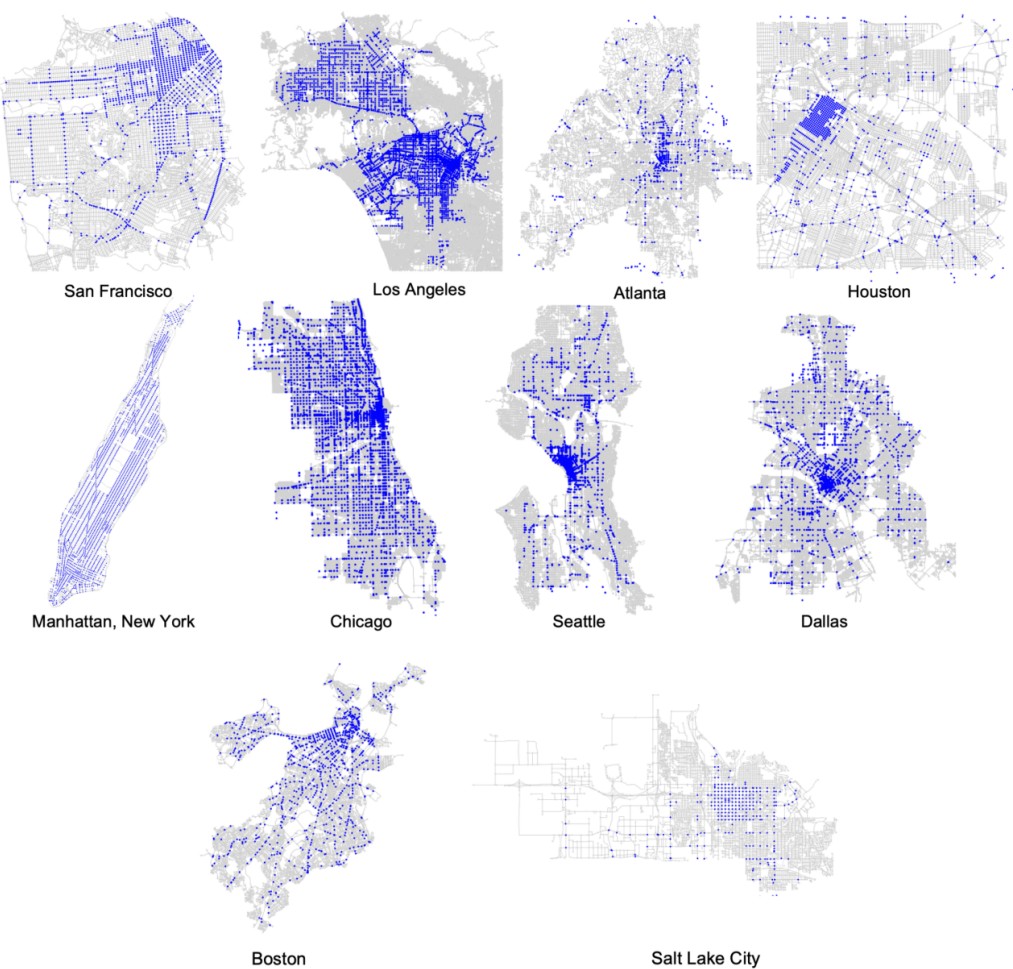

Figure 8: The distributions of the intersections considered in building the IntersectionZoo across the ten cities.

### A.3 EMISSION MODELS AND THEIR BEHAVIOR

A key requirement for capturing the effect of traffic scenarios on vehicle exhaust emission is a rich emission function. For this purpose, we integrate a suite of comprehensive and fast neural emission models (Sanchez et al., 2022) that replicate the industry-standard Motor Vehicle Emission Simulator (MOVES) (epa). For more details on how these neural surrogate emission models are built, we refer the readers to Sanchez et al. (Sanchez et al., 2022).

In Figure 9, we present a visual illustration of the emission landscape of some of these emission models when the vehicle's instantaneous acceleration and velocity are changed. We present these details for interested users as the objective of the CRL problem is specified by these emission models' behaviors. While different vehicle models under different conditions demonstrate varying emission quantities for a given velocity and acceleration, a few observations are generally common among them. First, decelerating under any velocity is preferred. This is consistent with the design of the GLOSA controller, which adopts a gliding behavior when the signal is red. Second, the higher the acceleration, the more the emission with higher velocity and higher acceleration pairs may result in the highest emissions.

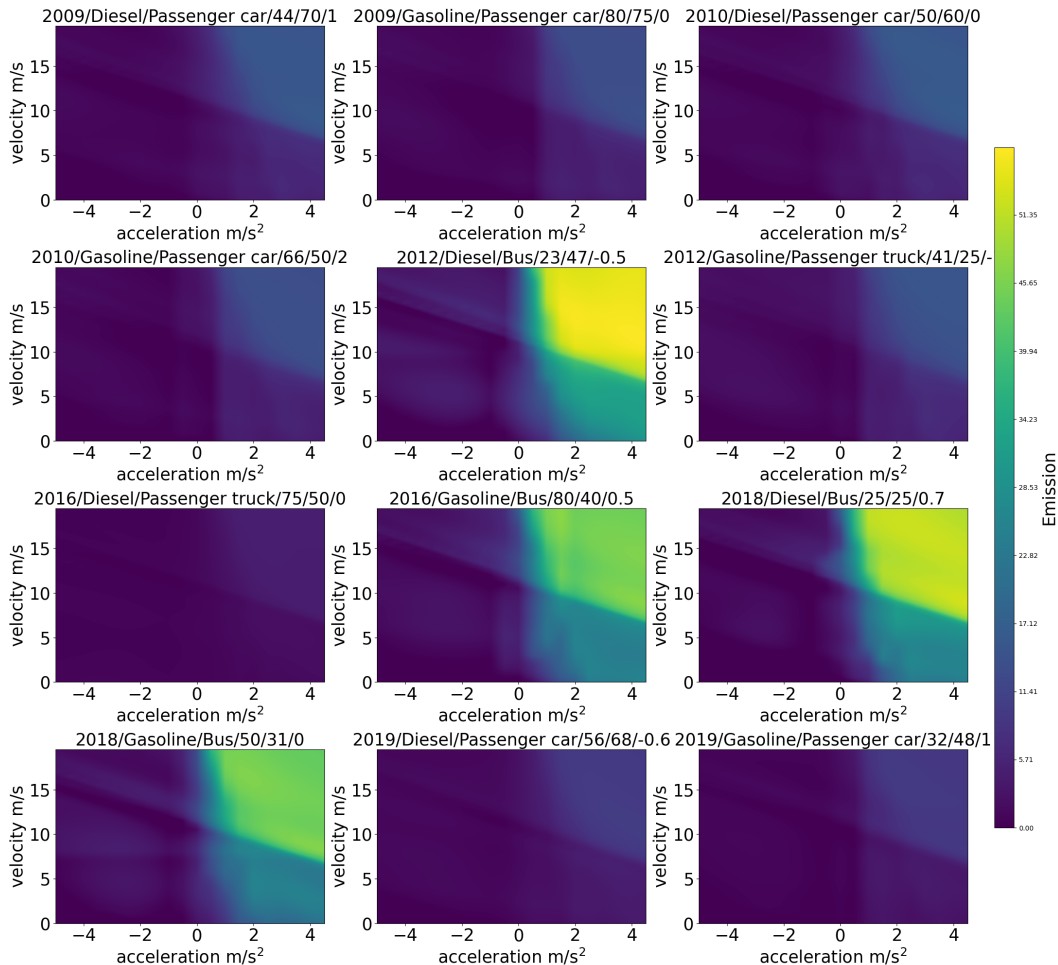

Figure 9: Emission landscapes of neural surrogate emission models. The figure depicts several randomly chosen contour plots depicting emission landscapes from various vehicle models operating under different conditions. Each plot title should be interpreted as *vehicle manufacture year/ fuel type/ vehicle type/ temperature/ humidity/ road grade*. Temperature is in Celsius, and humidity is the relative percentage, and the road grade is in degrees.

## A.4 DEFINING CONTEXT MDPs

**State Space**: The design of the observed state of a vehicle is mainly based on the capabilities of existing sensor technologies. The observed state includes the speed of the ego-vehicle, relative distance to the traffic signal, traffic signal state (red, green, or yellow) for the current phase, time remaining in the current phase, time remaining until the traffic signal turns green for the second and third cycle, vehicle location (i.e., a flag indicating whether the vehicle is approaching the intersection, at the intersection, or exiting the intersection), index of the vehicle's current lane, vehicle's intention to turn right, left, or go straight at the upcoming intersection, and for the follower and the leader vehicles on the same lane, adjacent right lane, and left lane of the ego-vehicle: speed, relative distance, turn signals status (turning right, left, or none).

As mentioned earlier, IntersectionZoo has both observed features and unobserved PCG-based features defining the context of a context-MDP. For users interested in conditioning the policies based on the context, we provide observable context features that include eco-driving adoption level, signal timing plan for the traffic signal phase relevant to the vehicle, atmospheric conditions such as temperature and humidity, the fuel type (electric or internal combustion engine), and information about the ego-vehicle's current approach (number of lanes, lane length, speed limit). The decision on which

features are available for conditioning is also based on the feasibility of implementing them in the real world.

**Action Space**: Longitudinal acceleration of each CV. For lane changing, a standard rule-based controller is used (Erdmann, 2014). This focuses IntersectionZoo on the continuous control aspect of eco-driving.

**Multi-objective Reward Function**: The reward $r_i^t$ for each CV $i$ at time $t$ is defined in Equation 10 inspired by Jayawardana et al. (2024). Here, $n_t$ is the vehicle fleet size, $v_t^i$ is the velocity, and $e_t^i$ is the $CO_2$ emissions of vehicle $i$ at time $t$. Hyperparameters include, $\eta$, $\alpha$, $\beta$, and $\tau$. The indicator function $\mathbb{1}_{v_t^i < \tau}$ indicates whether the vehicle is stopped, while the term $e_t^i$ encourages low emissions. The velocity term captures the effect on travel time. Users can configure the parameter $\eta$ to either get a fleet-based reward, agent-based reward, or a combination of both. All such formulations are acceptable.

$$r_t^i = \eta \frac{1}{n_t} \sum_{j=1}^{n_t} (v_t^j + \alpha \mathbb{1}_{v_t^j < \tau} + \beta e_t^j) + (1 - \eta)(v_t^i + \alpha \mathbb{1}_{v_t^i < \tau} + \beta e_t^i) \tag{10}$$

Users can extend the above default objective with additional reward terms as explained in Section A.6.

## A.5 Design rationale behind reward function and observation function

Our decisions on reward function design and observation design have been shaped by existing eco-driving literature and the existing sensor technologies to capture the state of a vehicle. Below we detail some of the decision choices and reasons behind them.

**Reward function**: As often discussed in eco-driving literature, many studies use emission and vehicle speed as the subterms of the objective function of the optimization methods (Jayawardana & Wu, 2022; Wegener et al., 2021; Zhu et al., 2023; Jayawardana et al., 2024). The emission term is a natural choice as the goal is to reduce the emission. The speed term is used as a shaped proxy reward for throughput maximization, as throughput itself would be a sparse reward (that can only be calculated at the end of the episode). Then, we introduce a term that penalizes stopping. This has also previously been used in eco-driving literature Jayawardana & Wu (2022). In our case, the choice to do this serves three purposes. First, based on the common emission model behaviors as explained in Section 9, decelerating is often a preferred action to reduce emission with stopping, and idling often emits more than decelerating. Therefore, it is preferred to encourage gliding behavior. This has been the core design principle of the GLOSA controller, and our penalty term encourages gliding to avoid idling. Second, if stopping is not penalized, methods could reward cheat by making the controlled vehicles stop at the beginning of an incoming approach, blocking the other vehicles and hence reducing emissions. With the stopping penalty term, we discourage these undesired behaviors. Third, the gliding behavior encouraged by the stopping penalty term also results in a smoother vehicle jerk, which could be more comfortable for the passengers. Therefore, we use these three terms in the reward. Following previous eco-driving work (Jayawardana & Wu, 2022; Wegener et al., 2021; Zhu et al., 2023; Jayawardana et al., 2024), we use a linear combination of these terms as the final objective, where weight coefficients are found by setting the weights based on each term's relative scale and doing a hyperparameter search around the starting configuration.

**Observation function**: Our observation function features are designed with the requirement that they should be available to the vehicle with the existing sensor technologies both onboard the vehicle and based on roadside units (RSU). By design, we provide all features that are considered in previous eco-driving literature. This is often the strategy adopted by existing benchmarking work based on real-world applications Ault & Sharon (2021); Vinitsky et al. (2022). The rationale behind this decision is to make sure that none of the essential features are missing from the observation, and if some features are less important for solving the problem, a given method could learn to ignore them. If required, as explained in Section A.8, users are allowed to perform observation shaping.

## A.6 Additional objectives terms and their encoding schemes

IntersectionZoo provides additional objective terms for users who wish to assess the effect of multiple objectives on generalization.

**Passenger comfort**: To accommodate passenger comfort, vehicles should maintain low accelerations and decelerations. To encourage this behavior, a reward term is defined as the $|a_t|$ where $a_t$ is the acceleration (or deceleration) of the vehicle at time $t$. When used with shared fleet-wise reward, the mean of $|a_t|$ across all vehicles is used.

**Kinematic realism**: Vehicles often cannot have high jerks (changes in accelerations in unit time) as actuators have jerk limits. To account for this, IntersectionZoo provides jerk control as $|a_t - a_{t-1}|$ where $a_t$ is the acceleration (or deceleration) of the vehicle at time $t$. When used with shared fleet-wise reward, the mean jerk across all vehicles is used.

**Fleet-level safety**: While individual vehicle safety is ensured using pre-defined rule-based checks, IntersectionZoo provide surrogate safety measures such as Time To Collision (TTC) to improve traffic flow level safety. These surrogate safety measures are commonly used by traffic engineers to measure the impact of new roadway interventions (Wang et al., 2021a).

Time to Collision (TTC) for a vehicle is measured as the time it would take for the vehicle to collide if it were to continue moving along its current paths without any changes in speed or direction. Formally, $TTC = \frac{\Delta d}{\Delta v}$ where $\Delta d$ is the relative distance and $\Delta v$ is the relative velocity. Both distance and velocity are measured relative to the leading vehicle of the ego-vehicle. In using TTC for fleet-level safety, we take the minimum TTC value across all vehicles at a given time step and share it with all vehicles.

### A.7 EVALUATION METRICS

To measure the generalization performance of a learned CRL policy, we leverage two metrics.

**Average emission benefits**: This measures the per vehicle per time step $CO_2$ emissions in grams based on the CRL policy and compares that with the human driving baseline. The lower the emissions, the better. Percentage emission reduction is given as the benefit.

**Average intersection throughput benefits**: This measures the number of vehicles that cross the intersection during an episode. We present it as the average throughput change percentage compared to the human driving baseline.

In addition to the task-specific metrics, users may present the training stability, sample complexity, and qualitative evaluations of the behavior of the agents through visualization as either supportive metrics or main evaluation metrics depending on the goals of the benchmarking. Furthermore, if additional objective terms such as passenger comfort and kinematic realism of vehicles are used in the objective as explained in Section A.6, it is important to present relevant metrics that can capture the effects of these objectives. As these metrics are often experimental purpose-specific, we do not provide a default implementation and leave it to the users to define their custom metrics.

### A.8 GUIDELINES FOR USING INTERSECTIONZOO

To standardize the use of IntersectionZoo, we provide the following guidelines.

- The environment comes with a default reward and observation function definitions. However, users are allowed to perform reward and observation shaping.

- A key requirement for assessing the performance is to ensure intersection throughput per intersection is never reduced by the learned eco-driving policy as explained in Section 2.2. If, for any intersection, the throughput is reduced (even if there are low emissions), its emission and throughput benefits must be set to zero.

- We recommend presenting the performance histograms (as given in Section 6) apart from presenting the average emission benefits over the intersection approaches of a given CMDP to visualize the generalization capacity of algorithms. This would facilitate identifying methods that overfit to a few specific intersections with dominating performances.

- The bounds on the controlled vehicle acceleration, minimum possible time to the vehicle in front, minimum headway, and willingness to obey the speed limit rules should be respected. These are set to constraint the actions based on vehicle actuator limits and to ensure safety.

### A.9 License details and accessibility

Our code and the IntersectionZoo are released under the MIT License and are available at https://github.com/mit-wu-lab/IntersectionZoo. The intersection datasets are also released under the MIT License. All data used for creating traffic scenarios are based on publicly available open data. SUMO traffic simulator is licensed under the EPL-2.0 with GPL v2 or later as a secondary license option (refer to SUMO website for more details).

## B APPENDIX B - EXPERIMENTAL SETUP AND RESULTS

### B.1 EXPERIMENTAL SETUP

All experiments are carried out using RLLib (Liang et al., 2018) with the default hyperparameter configuration. All policies are trained as multi-task learning policies where the context to condition the policy is as defined in Section A.4. We leverage 10 multiple workers in training the multi-task learning policies. Experiments were carried out in a computing cluster with 20 CPUs and an NVidia Volta V100 GPU with 32GB RAM. Each benchmarking run took roughly 24 hours in RLLib, with 5000 episodes (each with a horizon of 1000 steps with 50 warmups). We purposely ran each run for large number of iterations to ensure no further training can improve the policies. Benchmarking runs can be run for a shorter number of iterations, reducing computation times further.

### B.2 NOTES ON EVALUATIONS

For the reported results in Section 6, for each algorithm, we train with four random seeds. We train for 500 training iterations to ensure policies are well-converged. During the evaluation, we select the best-performing policy based on the rewards, vehicle throughout, and emission reductions.

### B.3 SALT LAKE CITY IN-DISTRIBUTION GENERALIZATION ANALYSIS

In this section, we demonstrate the emission benefit histograms of Salt Lake City CMDP with different MARL methods. Figure 10 illustrates the results. As discussed in Section 6 for Atlanta CMDP, SLC CMDP behaves similarly. While the effective emission benefits are positive for most MARL methods, some context-MDPs are not solved, as can be seen from the histograms. This serves as evidence of the fact that current MARL methods do not appear to easily solve the CRL problem in IntersectionZoo CMDPs and hence indicate that IntersectionZoo provides reasonable room for future CRL algorithmic improvements.

### B.4 SYSTEMATICITY AND PRODUCTIVITY IN GENERALIZATION

Below, as an example use of IntersectionZoo, we assess two targeted evaluations of generalization using IntersectionZoo, namely systematicity and productivity (Kirk et al., 2021).

**Systematicity in generalization**: Systematicity is generalization using systematic recombination of known knowledge (Kirk et al., 2021; Hupkes et al., 2020). To test this ability of DDPG and PPO, we leverage IntersectionZoo's capability to procedurally generate context-MDPs. Following Kirk et al. (2021), we first define a set of context features and their corresponding values as a set of uniform distributions (per feature). Then, we train policies by sampling feature values from each distribution to create context vectors. However, certain feature value combinations are never used during training. During testing, we only use the feature value combinations that were not used in training. This tests the algorithms' ability to systematically combine known knowledge to generalize.

We define in Table 4, the feature distributions for training, and in Table 5, the feature distribution for evaluation. During training, we sample intersections from Table 4 defined distribution but also not in Table 5 defined distribution. During evaluations, we only sample intersections from Table 5 defined distribution.

The resultant performance histogram is given in Figure 11a. While exact performance results are not the core focus of this experiment but rather to show the use IntersectionZoo in performing an analysis like this, we observe both DDPG and PPO fail to systematically generalize; baseline performs better in most cases.

**Productivity in generalization**: Productivity is when the learned policies generalization beyond seen training data (Kirk et al., 2021; Hupkes et al., 2020). To test this in PPO and DDPG, we perform an OOD evaluation by using a policy trained on SLC CMDP with zero-shot transfer to Atlanta CMDP. The resultant performance histogram is given in Figure 11b.

We would like to note that the primary focus of this experiment is not the exact performance results, but rather demonstrating how IntersectionZoo can be effectively used to conduct such an analysis. That said, even though the in-distribution performance analysis in Section 6 shows the DDPG policy

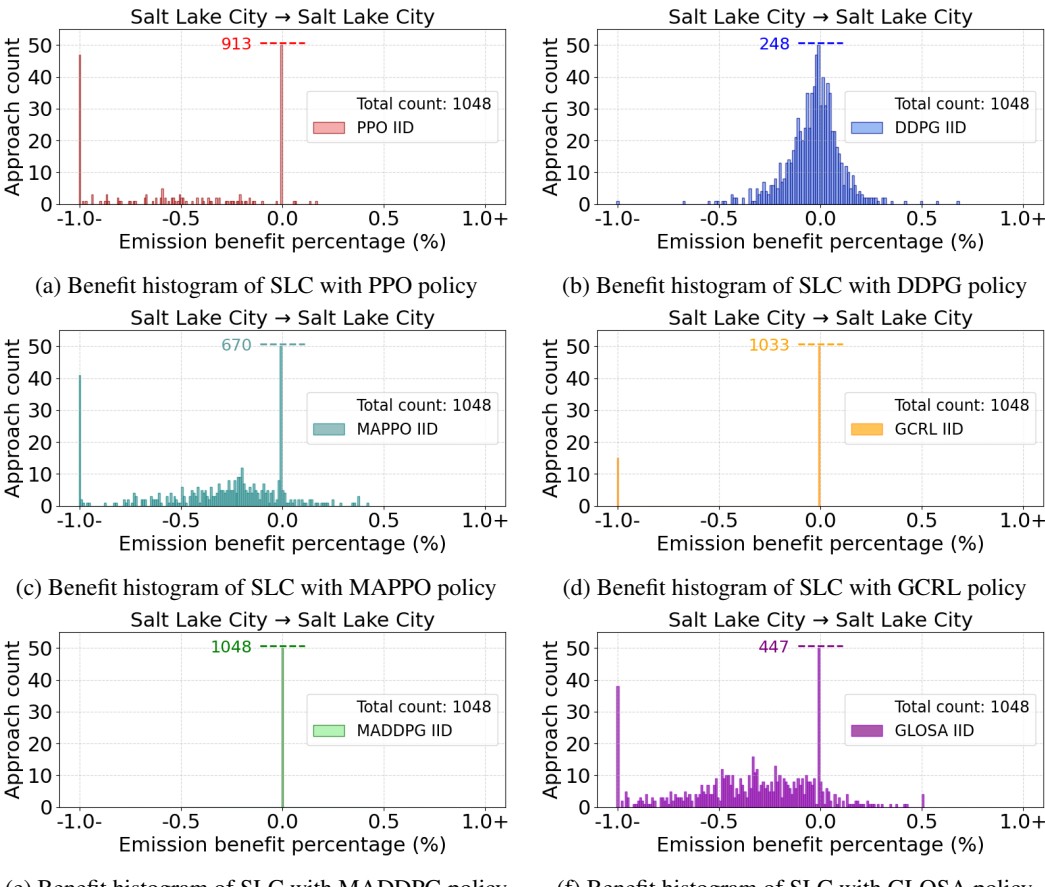

(a) Benefit histogram of SLC with PPO policy

(b) Benefit histogram of SLC with DDPG policy

(c) Benefit histogram of SLC with MAPPO policy

(d) Benefit histogram of SLC with GCRL policy

(e) Benefit histogram of SLC with MADDPG policy

(f) Benefit histogram of SLC with GLOSA policy

Figure 10: Emission benefit histograms of Salt Lake City (SLC) under different RL algorithms with in-distribution testing. Percentages are relative to the IDM baseline. Large y-axis counts are truncated for clarity. The total approach count refers to the total number of intersection approaches. The spikes at 0% are in part due to the zeroing of emissions benefits for any scenarios where throughput is reduced.

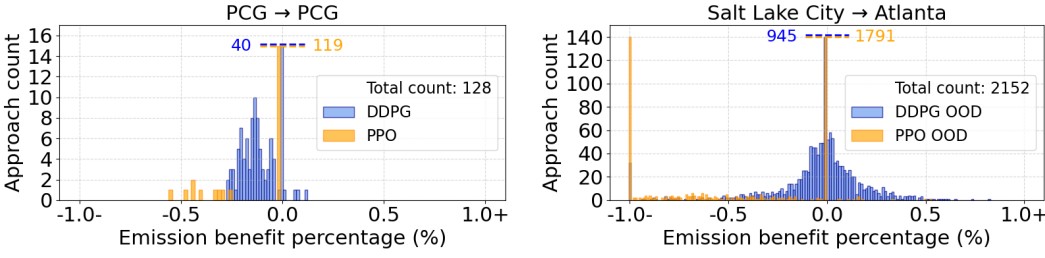

(a) Systematicity evaluation using procedurally generated intersections

(b) Productivity evaluation using policy transfer from Salt Lake City to Atlanta

Figure 11: Performance histograms of PPO and DDPG in assessing systematicity and productivity in generalization. All emission benefit percentages are measured relative to the human-driving baseline. For y-axis counts that are large, we truncate them for better visualization and indicate the count on the plot. The spikes at 0% are in part due to the aforementioned zeroing of emissions benefits for any scenarios where throughput is reduced. The total approach count is also given in each plot for reference, and the title indicates *train CMDP → test CMDP*.

trained on SLC CMDP seems to perform slightly better than the policy trained on Atlanta CMDP,

| Feature | Value range |
|---|---|
| Lane setup | (1,1), (1,2), (2,1), (2,2), (3,1), (3,2) - Format: (lane count, phase count) |
| Vehicle inflow | [100, 600] vehicles per hour |
| Green phase time | [20, 32] seconds |
| Red phase time | [20, 32] seconds |
| Lane length | [100, 775] meters |
| Speed limit | [16, 20] m/s |
| Signal offset | [1,3] seconds |

Table 4: Feature distribution for training for systematicity

| Feature | Value range |
|---|---|
| Lane setup | (2,2), (3,1) - Format: (lane count, phase count) |
| Vehicle inflow | [400, 500] vehicles per hour |
| Green phase time | [26, 29] seconds |
| Red phase time | [26, 29] seconds |
| Lane length | [325, 500] meters |
| Speed limit | [17, 18] m/s |
| Signal offset | [2, 3] seconds |

Table 5: Feature distribution for evaluation of systematicity

after the transfer, both DDPG and PPO seem to perform poorly, further indicating the limitations of existing RL algorithms when it comes to generalization across problem variations.

