# OpenReview forum: "IntersectionZoo: Eco-driving for Benchmarking Multi-Agent Contextual Reinforcement Learning"
_ICLR.cc/2025/Conference — ICLR 2025 Poster_

### Official Review · Reviewer_giey · 2024-10-28

**Soundness:** 3
**Presentation:** 3
**Contribution:** 3
**Rating:** 6
**Confidence:** 4

**Summary:**

This paper proposes IntersectionZoo, a comprehensive benchmark suite for multi-agent CRL through the real-world application of cooperative eco-driving in urban road networks. The authors ground IntersectionZoo in real-world applications, and capture real-world problem characteristics that are neglected in previous datasets, including partial observability, and multiple competing objectives. Also, IntersectionZoo covers 16,334 signalized intersections derived from 10 major US cities, enabling the testing of policy transfer between different cities. Using traffic scenarios in the proposed dataset, the authors benchmark various popular multi-agent RL and human-like driving algorithms and demonstrate their performance in Eco-driving.

**Strengths:**

1. This paper focus on benchmarking the task of cooperative Eco-driving. The idea of this task is to control a fleet of vehicles to reduce fleet-level vehicular emissions. The task sounds interesting and is absolutely beneficial to reduce air pollution in the era of global climate change.
2. The authors ground IntersectionZoo in real-world applications, and capture real-world problem characteristics that are neglected in previous datasets, including partial observability, and multiple competing objectives.
3. The author model a wide range of major factors affecting vehicular exhaust emissions, including temperature, road conditions, travel demand, making the traffic scenarios provided in IntersectionZoo more realistic and reliable.
4. The dataset is of substantial volume, including one million scenarios simulated in 16,334 signalized intersections derived from 10 major US cities. This enable robust testing results of a given algorithm.

**Weaknesses:**

1. In this paper, the framework of IntersectionZoo looks sophisticated, but there lacks experimental results that illustrate how precise the simulation results are. Can they reflect the real-word vehicular exhaust emissions precisely?
2. When benchmarking popular algorithms with IntersectionZoo, the authors show the performance of some algorithms from various aspects. However, what is the rationality of these results? How do they strengthen the contribution of the proposed dataset?
3. The authors test the performance of some algorithms from various aspects with IntersectionZoo, but there lacks the testing results with other comparable benchmarks (like the ones in Table 1). How do the results on other benchmarks differ from the results on IntersectionZoo? Will the differences illustrate the advantage of IntersectionZoo?

**Questions:**

1. Could you please provide some experimental results to verify that IntersectionZoo can reflect the real-word vehicular exhaust emissions precisely?
2. Could you please explain the rationality of your benchmarking results? How do they strengthen the contribution of the proposed dataset?
3. Could you please provide some major testing results of the popular algorithms on other existing benchmarks? How do the results on other comparable benchmarks differ from the results on IntersectionZoo? Will the differences illustrate the advantage of IntersectionZoo?

---

> ### Author Response · Authors · 2024-11-22
> **Rebuttal by Authors (1/2)**
>
> We are thankful to the reviewer for their feedback and suggestion on our work. We are glad that reviewer liked the impact of the work. Below we provide explanations and answers to reviewer’s questions and concerns.
>
> - In this paper, the framework of IntersectionZoo looks sophisticated, but there lacks experimental results that illustrate how precise the simulation results are. Can they reflect the real-word vehicular exhaust emissions precisely?
> - Could you please provide some experimental results to verify that IntersectionZoo can reflect the real-word vehicular exhaust emissions precisely?
>
>
> We ensure IntersectionZoo reflects real-world conditions by using real-world data and calibrated simulations (see Section 4.1 and Appendix A). For vehicle emissions, we use neural surrogates of the industry-standard MOVES model [1] by the U.S. Environmental Protection Agency. The MOVES model has been validated with real-world data by the EPA and is the standard vehicle exhaust emission model used in the industry. By using it, we make sure IntersectionZoo also reflects the real-world emission values. Detailed analysis of these emission models is provided in Appendix A.4.
>
> [1] https://www.epa.gov/moves/latest-version-motor-vehicle-emission-simulator-moves
>
>
> - When benchmarking popular algorithms with IntersectionZoo, the authors show the performance of some algorithms from various aspects. However, what is the rationality of these results? How do they strengthen the contribution of the proposed dataset?
> - Could you please explain the rationality of your benchmarking results? How do they strengthen the contribution of the proposed dataset?
>
> This is a great question!! This question from the reviewer led to some insightful discussions among the authors and for that we are grateful. Given the important of answering this question, we provided a general response with the title “Goals of experiments” under the section “More analysis on benchmarking”. We kindly request the reviewer to refer to that section in the general response section for more details.
>
> Following on the reviewer’s comment “How do they strengthen the contribution of the proposed dataset?”, we also want to highlight and clarify that what we propose in this work is not just a dataset but a comprehensive suite of dataset, simulator, evaluation metrics and training frameworks to benchmark contextual multi-agent RL. The dataset of real-world intersections is only a one part of the whole framework.

---

> ### Author Response · Authors · 2024-11-22
> **Rebuttall by Authors (2/2)**
>
> - The authors test the performance of some algorithms from various aspects with IntersectionZoo, but there lacks the testing results with other comparable benchmarks (like the ones in Table 1). How do the results on other benchmarks differ from the results on IntersectionZoo? Will the differences illustrate the advantage of IntersectionZoo?
> - Could you please provide some major testing results of the popular algorithms on other existing benchmarks? How do the results on other comparable benchmarks differ from the results on IntersectionZoo? Will the differences illustrate the advantage of IntersectionZoo?
>
>
> Thank you for the suggestion. We developed IntersectionZoo because none of the benchmarks in Table 1 offer its unique features. Our comparison with 19 other RL benchmarks across 11 characteristics highlights this gap. Specifically, no existing benchmark enables contextual multi-agent RL benchmarking grounded in real-world context distributions. As noted in the related work section, CRL benchmarks not tied to real-world tasks risk exploitable structures that hinder progress. For example, in the SMAC benchmark, policies based solely on timestep data can achieve high win rates [1]. As such, benchmarking a new method on the existing benchmarks (Table 1) can serve to validate the method, but benchmarking on IntersectionZoo can provide added evidence that the method will translate to real-world applications. This research gap is the opportunity IntersectionZoo fills in and is the advantage of IntersectionZoo over the other existing benchmarks.
>
> Having said this, we also believe there should be more than one benchmarks when assessing the performance of methods as diverse tasks make the bechmarking more rigorous. Hence having different benchmark tasks for the same purpose is infact an important requirement. This is similar to what happens in single agent RL where, for example, Mujoco and Gym classic control tasks co-exists complementing each other. The co-existence of more than one benchmark for the same purpose is important for the progress in the field. Therefore, we hope following the trend set by introduction of IntersctionZoo, there will be more benchmarks that focus on contextual MARL.  We hope this answer clarifies the question the reviewer has.
>
> [1] Ellis at el. Smacv2: An improved benchmark for cooperative multi- agent reinforcement learning.
>
>
> We hope our rebuttal sufficiently addresses the concerns that the reviewer has raised. Therefore, we request the reviewer to consider increasing our score if these presented results are satisfying and address the reviewer’s concerns and suggestions. We thank the reviewer for taking time to review our work!

---

> > ### Author Response · Authors · 2024-11-25
> > **Check in**
> >
> > Hi reviewer giey,
> >
> > We sincerely appreciate your thoughtful feedback, which helped us strengthen our work and improve the clarity of the benchmarking section. With the rebuttal period ending tomorrow, we wanted to check if our responses have adequately addressed the reviewers’ concerns. Please let us know if there are any remaining questions or if further clarification would be helpful to address your concerns. Thank you once again for your time and effort in reviewing our submission.

---

> ### Comment · Reviewer_giey · 2024-11-26
>
> Thanks for your response. My major concerns have been solved, and I would like to raise my score to 6.

---

> > ### Author Response · Authors · 2024-11-26
> > **Thank you!**
> >
> > Thank you for taking the time to review our submission and check our rebuttal. We’re pleased to hear that it addressed your concerns. We sincerely appreciate your thoughtful feedback and thank you for the updated score!

---

### Official Review · Reviewer_r4Ef · 2024-11-01

**Soundness:** 3
**Presentation:** 4
**Contribution:** 4
**Rating:** 6
**Confidence:** 3

**Summary:**

The paper proposes a benchmark suite, IntersectionZoo, for contextual reinforcement learning evaluation. The suite is designed based on a complex real-world task, i.e., cooperative eco-driving at signalized intersections. This paper well summarizes the recent work related to RL benchmark. Several experiments are conducted to test some popular multi-agent RL algorithms and show that all the adopted algorithms perform poorly on generalization. The open-source code is also provided to facilitate the community.

**Strengths:**

The originality is good. This paper addresses the problem of contextual reinforcement learning by creating a comprehensive benchmark for eco-driving simulation.

The problem formulation is presented clearly and in detail.

The literature review offers an extensive and in-depth overview of related work.

The proposed IntersectionZoo benchmark can be a boost for contexture RL research.

Open-source code is provided to support the research community.

**Weaknesses:**

* The main content seems to exceed the page limit – the 8. The Reproducibility Statement is on the 11th page.

* PPO and DDPG are not specifically for multi-agent RL but are only used with independent agents. It seems not reasonable to choose these two algorithms for testing. Besides, for systematicity and productivity in generalization tests, the authors only use PPO and DDPG for evaluation, which is confusing.

* From your experiments, it’s hard to get the conclusion in Line 532 that “the popular multi- agent RL algorithms struggle to generalize in CRL settings” since only two MARL-based methods, i.e., MAPPO and GCRL, are adopted for evaluation, and just on the in-distribution generalization.

* Given the unsatisfactory results of the chosen multi-agent RL algorithms on this benchmark, it would be better to briefly review the latest multi-agent RL to justify why you chose PPO, DDPG, MAPPO, and GCRL for evaluation.

**Questions:**

What does the “baseline” refer to in “Both DDPG and PPO fail to systematically generalize; baseline performs better in almost all cases” in Line 497?

---

> ### Author Response · Authors · 2024-11-22
> **Rebuttal by Authors**
>
> We’re grateful for the feedback and suggestions of the reviewer. It helps us strengthen the work and rethink about how we should best present the experimental results. Here, we provide explanations addressing reviewer’s concerns and suggestions.
>
> - The main content seems to exceed the page limit – the 8. The Reproducibility Statement is on the 11th page.
>
> ICLR author guidelines for 2025 states that the page limit is 10 and reproducibility statement does not count for page limit.
>
> - PPO and DDPG are not specifically for multi-agent RL but are only used with independent agents. It seems not reasonable to choose these two algorithms for testing. Besides, for systematicity and productivity in generalization tests, the authors only use PPO and DDPG for evaluation, which is confusing.
>
> We thank the reviewer for the suggestion. While our work is focused on multi-agent control, it specifically focuses on contextual multi-agent RL. Given this, we believe PPO and DDPG as independent agents are worth exploring given their perhaps better success in generalization. Furthermore, PPO has shown to outperform many of the other MARL algorithms indicating even if it is about independent agents, such methods are competitive in the multi-agent setting [1]. Also, many of the MARL libraries provide independent PPO and DDPG as multi-agent control methods. Given these reasons, we believe they are justified to be used in this work.
>
> While we use PPO and DDPG, we also use their counterpart multi-agent versions for benchmarking as well. We have three MARL methods: MAPPO, MADDPG and GCRL. Therefore, we think we cover all fronts with our analysis.
>
> Our goals with systematicity and productivity analyses is to use as examples to show how IntersectionZoo can be used to perform such an analysis. These are given as example use cases and hence, we did not exhaustively analyze all methods.
>
> [1] Yu et al. The surprising effectiveness of ppo in cooperative multi-agent games. 2022
>
> [2] https://marllib.readthedocs.io/en/latest/
>
>
> - From your experiments, it’s hard to get the conclusion in Line 532 that “the popular multi-agent RL algorithms struggle to generalize in CRL settings” since only two MARL-based methods, i.e., MAPPO and GCRL, are adopted for evaluation, and just on the in-distribution generalization.
>
> We thank the reviewer for raising this point. Taking the reviewer’s suggestion, we also added MADDPG as a MARL method to the analysis. While DDPG works relatively well in Atlanta CMDP, it fails to generalize in Salt Lake City CMDP. Thus, we believe the statement we made in line 532 is valid statement. Having said that, given our focus on muti-agent CRL as explained earlier, we believe PPO and DDPG still counts when analyzing and making observations and they also show similar observation further strengthening the claim.
>
> We also note that we changed the Section 6 of the paper to revise the goals of the experiments and summarized these changes in the general response section titled “More analysis on benchmarking”. We kindly request the reviewer to refer to that section for more details.
>
> - Given the unsatisfactory results of the chosen multi-agent RL algorithms on this benchmark, it would be better to briefly review the latest multi-agent RL to justify why you chose PPO, DDPG, MAPPO, and GCRL for evaluation.
>
> We thank the reviewer for the suggestion to discuss the thoughts behind the choices of the algorithms. First, we would like to mention that based on reviewer’s suggestions, we added MADDPG as another MARL method and GLOSA controller as a rule-based method to further strengthen the benchmarking section. We refer the reviewer to general response section titled “More analysis on benchmarking” for more details on this.
>
> We list the decision rationale for the choices for PPO, DDPG in the previous answer and hence omit the details here. For MAPPO, MADDPG and GCRL, we used them as they are popular and established multi-agent RL methods.
>
> - What does the “baseline” refer to in “Both DDPG and PPO fail to systematically generalize; baseline performs better in almost all cases” in Line 497?
>
> It refers to the human-like driving behaviors given by IDM method. To make this more clear, we added a separate section, 5.2, describing the baselines.
>
> Finally, we are grateful to the reviewer for their feedback and suggestions on our work. We would like to request the reviewer to consider increasing the score if our additional experiment, analysis and our rebuttal address the reviewer's concerns and questions. We thank the reviewer for taking the time to review our work.

---

> ### Author Response · Authors · 2024-11-25
> **Check in**
>
> Hi reviewer r4Ef,
>
> Thank you again for your insightful suggestions, which have greatly improved our work and made the benchmarking section better with more baselines and discussions. As the rebuttal period comes to a close tomorrow, we wanted to check if our responses have sufficiently addressed the reviewers’ concerns. Please let us know if you have any further questions or if any clarification would be helpful. We sincerely appreciate the time and effort you’ve put into reviewing our submission. Thank you!

---

> > ### Comment · Reviewer_r4Ef · 2024-11-27
> > **Official Comment**
> >
> > Thank you for your response. After carefully reviewing it again, I will keep my score.

---

> > > ### Author Response · Authors · 2024-11-27
> > > **Thank you!**
> > >
> > > We thank the reviewer for their great suggestions and thoughts on how to improve our work. It definitely helped us strengthen the work. We hope our rebuttal was responsive to the reviewer's concerns on all fronts. If you have any further questions or thoughts, please let us know. Thanks again for taking the time to carefully review our submission and check our rebuttal. We appreciate it!

---

### Official Review · Reviewer_Ld56 · 2024-11-03

**Soundness:** 2
**Presentation:** 3
**Contribution:** 2
**Rating:** 6
**Confidence:** 3

**Summary:**

In this paper the authors introduce a new benchmark for multi-agent reinforcement learning (MARL) called IntersectionZoo, designed to address the cooperative eco-driving task within urban networks. The objective of cooperative eco-driving is to control a fleet of vehicles at signalized intersections to minimize fleet-wide emissions with minimal impact on individual travel times.

The authors discuss the benchmark in the context of Multi-Agent Contextual Reinforcement Learning (MA-CRL) and highlight how IntersectionZoo is purposefully tailored to generate multiple scenario variations (i.e., contexts) and evaluate the generalization capabilities of MA-CRL agents across such variations. Specifically, IntersectionZoo addresses a range of contextual factors, including intersection configuration, vehicle type distribution, fuel type distribution, human driver models, weather conditions, eco-driving adoption rates, peak and off-peak traffic patterns, among others. The benchmark leverages data across 10 cities and over 16,000 signalized intersections and is built on open-source software, e.g. SUMO for microscopic traffic modeling, and RLLib for RL training.

**Strengths:**

I resonate with the overall purpose of this work: bridging the gap between RL research (focused on benchmarks and generality of algorithms) and impactful, real-world applications. The paper is generally well-written and straightforward to follow. The authors thoroughly compare their benchmark with a comprehensive body of literature in the field. Based on my understanding, this work entails significant data collection and processing, which could provide substantial benefits to the broader transportation research community.

**Weaknesses:**

In my opinion, the primary limitation of this work is its relatively narrow contribution to the broader RL, or specifically multi-agent RL, community. Although the authors claim that IntersectionZoo aims to serve as a novel standardized benchmark for MA-CRL research, many of the benchmark’s design decisions and characteristics are highly tailored to cooperative eco-driving and to the specific problem formulation adopted within this work. While evaluating generalization performance across different scenario characteristics is important for this application, it seems questionable to assert that results and insights on IntersectionZoo would extend to MA-RL research more broadly. This, to me, somewhat undercuts the fundamental purpose of a benchmark.

For instance, while scenario factors vary, the overall application and formulation remain fixed. This impacts algorithmic choices as well, such as the mandatory choice of a continuous action space (acceleration of the controlled vehicles), a specific reward function design, limited environment scale (restricted to single intersections), predefined driver models, and no significant uncertainty sources. These are just a few examples.

To the best of my understanding, this work would have greater impact if it concentrated more on the application side—namely, contributing novel algorithms to address a critical problem—rather than positioning this benchmark to generate deep insights into MA-RL algorithm development.

Furthermore, the experimental analyses in this work need _considerable_ improvement. The authors provide a brief comparison of PPO, DDPG, MAPPO, and GCRL, concluding that these algorithms fail to generalize and perform poorly across all tested conditions. While experimental evaluation may not be the core focus of this paper, a thorough and insightful assessment of established MA-RL algorithms is essential in order to evaluate the usefulness of a benchmark. Is it the case that no algorithm performs adequately in any of the examined scenarios? The authors should consider expanding the experimental section by exhaustively benchmarking popular RL algorithms on the proposed benchmark, with in-depth evaluation and analyses.

**Questions:**

- Can the authors elaborate on how IntersectionZoo is expected to benefit the broader MA-RL community? What features of the cooperative eco-driving task are common to other MA-RL tasks? To what extent the insights obtained on IntersectionZoo can be extended to broader MA-RL settings? More specifically, could the authors suggest particular MA-RL domains where insights from IntersectionZoo are likely, or unlikely, to transfer? Providing both positive and negative examples would help the readers with assessing whether IntersectionZoo could serve as a relevant benchmark for their own purposes.

- Can the authors provide a more extensive experimental analysis of MA-RL algorithms on this benchmark? If no algorithm performs well on the benchmark, what are the underlying causes for such a performance gap? For example, IntersectionZoo focuses on two performance metrics: average emission benefits and average intersection throughput benefits. While these metrics are interesting from a domain-driven perspective, the paper would benefit from considering additional algorithm-driven metrics relating to, e.g., sample efficiency, stability of training, etc. Moreover, a qualitative visualization of the resulting policy would further improve the experimental assessment.

---

> ### Author Response · Authors · 2024-11-22
> **Rebuttal by Authors (1/3)**
>
> We thank the reviewer for their feedback on our work and the suggestions. It definitely has helped us strengthen the work better. Below we provide explanations and answers to the reviewer’s concerns and suggestions by grouping the reviewer comments when appropriate.
>
> - In my opinion, the primary limitation of this work is its relatively narrow contribution to the broader RL, or specifically multi-agent RL, community. Although the authors claim that IntersectionZoo aims to serve as a novel standardized benchmark for MA-CRL research, many of the benchmark’s design decisions and characteristics are highly tailored to cooperative eco-driving and to the specific problem formulation adopted within this work. While evaluating generalization performance across different scenario characteristics is important for this application, it seems questionable to assert that results and insights on IntersectionZoo would extend to MA-RL research more broadly. This, to me, somewhat undercuts the fundamental purpose of a benchmark.
> - Can the authors elaborate on how IntersectionZoo is expected to benefit the broader MA-RL community? What features of the cooperative eco-driving task are common to other MA-RL tasks? To what extent the insights obtained on IntersectionZoo can be extended to broader MA-RL settings? More specifically, could the authors suggest particular MA-RL domains where insights from IntersectionZoo are likely, or unlikely, to transfer? Providing both positive and negative examples would help the readers with assessing whether IntersectionZoo could serve as a relevant benchmark for their own purposes.
>
> We thank the reviewer for pointing out this concern and appreciate the reviewer’s thoughts on the topic. We thought to first share what made us work on this benchmark as we believe that may provide a context for why a benchmark like IntersectionZoo is necessary for the MARL community.
>
> In our experience studying and applying MARL (and the experience of many researchers we have spoken with), MARL algorithms remain surprisingly brittle to small changes in the environment. This is now a known phenomenon even for standard RL benchmark tasks [1,5]. While our proposed benchmark appears narrow (specific to the cooperative eco-driving task), we believe this is a reflection of the state of RL algorithms. We wish to contribute a benchmark that is richer than toy tasks (e.g. CARL [1]) and appropriate to the MARL community. While generalization across tasks is an eventual goal of MARL research, we believe it is meaningful to first address generalization within tasks, and we believe that IntersectionZoo would facilitate research in this direction.
>
> Having said that, IntersectionZoo is designed to capture this notion of problem variations that are prevalent in many real-world multi-agent problems. In doing so, we use eco-driving-based problem variations as variations arise naturally due to a given task, and we cannot isolate the variations from the tasks and attempt to provide general variations that are common to any task. We expect that the use of IntersectionZoo with any future MARL methods will show those algorithms’ generalization capacity across problem variations. This is what IntersectionZoo provides as an insight to the broader RL/MARL community. Further, by capturing factors that affect generalization performance, such as the impact of multiple objectives, we further enable the assessment of MARL algorithms on more than one evaluation criterion. By grounding IntersectionZoo in a real-world application of eco-driving, we aim to capture real-world problem variations and other challenges with two goals. 1) it captures the notion of the real-world difficulty of the problem without superficially making the problem easier or harder (the real level of difficulty introduced by problem variations), and 2) it allows targeted evaluations of generalizations (e.g., the impact of multiple objectives on contextual MARL). Therefore, the reason to make the IntersectionZoo specific about eco-driving is an intentional decision that actually strengthens the work. We also note that similar motivations are used in other benchmarks for single-agent CRL, which has progressed better than multi-agent CRL.  We hope following the introduction of IntersctionZoo and the motivation we elaborated here based on the emerging needs of real-world multi-agent RL, there will be more benchmarks that focus on contextual MARL in the future, and we are glad that we are to first fill this important yet remained to be filled research gap in the RL community.  We hope this answer clarifies the question the reviewer has.

---

> ### Author Response · Authors · 2024-11-22
> **Author rebuttal (2/3)**
>
> [1] Benjamins et al. Contextualize me–the case for context in reinforcement learning.
>
> [2] James et al. Rlbench: The robot learning benchmark & learning environment.
>
> [3] Lowe at al. Multi-agent actor-critic for mixed cooperative-competitive environments.
>
> [4] Wu et al. Flow: A modular learning framework for mixed autonomy traffic.
>
> [5] Jayawardana et al. The impact of task underspecification in evaluating deep reinforcement learning.
>
>
> - While scenario factors vary, the overall application and formulation remain fixed. This impacts algorithmic choices as well, such as the mandatory choice of a continuous action space (acceleration of the controlled vehicles), a specific reward function design, limited environment scale (restricted to single intersections), predefined driver models, and no significant uncertainty sources. These are just a few examples.
>
> We thank the reviewer for their thoughts on these points. We provide an explanation on each of those points below.
>
> **Action Space**: IntersectionZoo uses a continuous action space by default, with an option for discrete actions (e.g., lane changes). Users can configure experiments in the codebase.
>
> **Reward Function**: The reward function is flexible and customizable. A default function is provided to standardize use, with detailed rationale in Appendix A.5.
>
> **Environment Scale**: IntersectionZoo includes over 16,000 intersections and nearly 1 million traffic scenarios (context-MDPs), sufficient for contextual MARL benchmark. We provide the option for an interested user to define environments as more than one intersection and add them as a new dataset.
>
> **Predefined Driver Models**: Predefined human driver models simplify and standardize the benchmark. Users can add custom models if required, and the options are given in the code.
>
> **Stochasticity**: Stochasticity is built-in via randomized vehicle arrivals, CV order, and driver behaviors. Additional noise (e.g., imperfect action execution) can be configured in the codebase.
>
>
> - To the best of my understanding, this work would have greater impact if it concentrated more on the application side—namely, contributing novel algorithms to address a critical problem—rather than positioning this benchmark to generate deep insights into MA-RL algorithm development.
>
> We thank the reviewer for their suggestion on an alternative focus. We are definitely interested in the application side of this problem, as alluded to in our broader societal impact statement. One of the aims of this benchmark is exactly to facilitate the ML community to help researchers on the application side. Right now, it is not clear how to answer the application question without first (or at least making headway) on the fundamental question of generalization within tasks (contextual RL).
>
> Therefore, our goal is by giving this benchmark to RL community as a tool for assessing the generalization in MARL, we will in turn get algorithms that can solve eco-driving problem.  Plus, if someone is interested in using IntersectionZoo for eco-driving research, they can use IntersectionZoo directly and we highly encourage that as well.

---

> > ### Author Response · Authors · 2024-11-22
> > **Author rebuttal (3/3)**
> >
> > - Furthermore, the experimental analyses in this work need considerable improvement. The authors provide a brief comparison of PPO, DDPG, MAPPO, and GCRL, concluding that these algorithms fail to generalize and perform poorly across all tested conditions. While experimental evaluation may not be the core focus of this paper, a thorough and insightful assessment of established MA-RL algorithms is essential in order to evaluate the usefulness of a benchmark. Is it the case that no algorithm performs adequately in any of the examined scenarios? The authors should consider expanding the experimental section by exhaustively benchmarking popular RL algorithms on the proposed benchmark, with in-depth evaluation and analyses.
> > - Can the authors provide a more extensive experimental analysis of MA-RL algorithms on this benchmark? If no algorithm performs well on the benchmark, what are the underlying causes for such a performance gap?
> >
> > We thank the reviewer for the suggestion. Following the reviewer’s comment, we added MADDPG as an additional MARL method and GLOSA as a rule-based controller. With these additions we now have two baselines controllers and 5 MARL methods. We detail these implementations and results in the general response section titled “Benchmarking analysis with more methods”. We kindly request the reviewer to refer to that section for more details. We also revised the experiments section to articulate the our goals behind the experiments more clearly. This is also explained in the general response.
> >
> > Finally, we are grateful to the reviewer for their suggestions on multiple avenues of our work. We hope our rebuttal sufficiently addresses the concerns that the reviewer has raised. Therefore, we would like to request the reviewer to consider increasing our score if these presented results are satisfying and address the reviewer’s concerns and suggestions.

---

> ### Author Response · Authors · 2024-11-25
> **Check in**
>
> Hi reviewer Ld56,
>
> Thank you once again for your valuable suggestions, particularly regarding potential alternative focuses. Your feedback was very helpful for us to clarify the unique need for IntersectionZoo within the MARL community. As the rebuttal period is concluding tomorrow, we wanted to follow up to ensure our responses have adequately addressed the reviewers' concerns. Please let us know if there are any remaining questions or areas where further clarification would be helpful. We appreciate the time and effort you've dedicated to reviewing our work. Thank you!

---

> > ### Comment · Reviewer_Ld56 · 2024-11-28
> > **Thank you for your response**
> >
> > I'd like to thank the authors for their thoughtful response. I believe the quality of this work has improved as a result of the rebuttal (e.g., thanks to the inclusion of additional baselines) and I'll be raising my score to 6.

---

> > > ### Author Response · Authors · 2024-11-28
> > > **Thank you!**
> > >
> > > We want to thank the reviewer for their thoughtful review, which made us rethink and clarify the unique need for IntersectionZoo within the CRL/MARL research community. We are pleased that the reviewer found the quality of the work has improved with our rebuttal. We thank the reviewer for raising the score and for taking the time to review our work and check the rebuttal. We appreciate it!

---

### Official Review · Reviewer_dDfH · 2024-11-05

**Soundness:** 3
**Presentation:** 2
**Contribution:** 3
**Rating:** 6
**Confidence:** 5

**Summary:**

**Overview**
This work proposes a new benchmark for testing the generalizability of MARL algorithms across different MDP settings. With the background of reducing $CO_2$ emission by controlling vehicle fleets, progress on this benchmark may potentially generate real-world impact and advance the development of MRL.

**Method**
Specifically, intersections were collected for 10 major cities in the US and reconstructed in the SUMO simulator with a set of configurable parameters (context) like vehicle throughput, speed limit, temperature, the ratio of autonomous cars, and so on. The parameter distribution is provided by referring to literature from the transportation department. Thus, sampling parameters from these distributions allow the simulator to build intersection scenarios similar to its real-world counterpart. Each parameter combination forms a special MDP, and thus training agents with scenarios sampled from the same distribution allows benchmarking in-distribution performance. Testing agents in scenarios built with another parameter distribution can reflect the algorithm's OOD performance.

**Experiment**
The experiment shows that the current SOTA MARL method struggles to do even in-distribution generalization, indicating there is a large room for improvement.

**Strengths:**

1. It is a good application for MARL researchers to study how to build algorithms to further reduce exhaust emissions and make a real-world impact.
2. The scenarios database is large. With realistic context distribution, IntersectionZoo is a good place to study the generalizability of algorithms. Also, the whole system including the intersection data, simulator, metric, and training framework is built reasonably and solidly.
3. As an open-sourced benchmark, the code is of good quality with documentation included. Seems it is ready for use.

**Weaknesses:**

1. The writing is poor. The experiment figures are hard to understand. Specifically, what does the total count mean? Though I can roughly infer it, it would be better to explain it in the paper. For the benefit percentage, does 0.5 means 0.5% or 50%. If it is 0.5%, it seems the emission doesn't change a lot by switching to MARL control, and the key is that MARL agents can not achieve the same level of throughput as human drivers. However, it seems GCRL agent is close to human drivers' performance. I suggest giving more analysis and explanation to these figures. It can be done in the captions since I found the captions of Figure 5 and Figure 6 are simple copy-paste. Also, is it possible to provide more visualizations in the appendix or even provide a video? So readers can know how different the map topologies and vehicle distributions are for these cities. I can not imagine a large dataset from pure text.
2. More experiments or results are needed. Though the experiments show that MARL can not generalize well, we don't know if it is due to the algorithm itself or if the task is too difficult to learn for RL agents, i.e. improper reward function and observation or MDP definition. One way to clarify this is to provide training performance as well. It needs to be proved first that the MARL can overfit the training scenarios and exceed human driver performance. Then we can confirm that the simulation design is appropriate, so MARL agents have the chance to surpass humans, and the failure on the test set is indeed due to poor generalizability. Also, parameters are fixed for the experiments like the adoption rate, which is 1/3. Did the authors conduct experiments to see which factor affects the performance most? I believe the adoption rate is an important factor since the MARL performance would be largely influenced by the population size. Also, a baseline can be added to use the IDM vehicle control these CVs to see if the MARL can outperform this simple rule-based baseline at least at training time. With such a trend, we can conclude it shows some signs that MARL can achieve better performance, and thus work on this benchmark is promising.

**Questions:**

1. How would the temperature and humidity affect the policy behavior? For example, is it better to drive at low speed in summer to reduce the CO2 emission? But it would reduce the throughput right?
2. As far as I know, the OSM data only provides the location of the traffic light and doesn't tell which lane it controlls. How does this information be completed in IntersectionZoo?
3. Type: Appendix, table 3. evaluation -> training

---

> ### Author Response · Authors · 2024-11-22
> **Rebuttal by Authors (1/3)**
>
> We thank the reviewer for a well-thought review on the work. The suggestions of the reviewer definitely helped us to strengthen the work better. We are glad that reviewer found our work to be impactful for MARL research. Below, we address the reviewer's suggestions and concerns.
>
> - The experiment figures are hard to understand. Specifically, what does the total count mean? Though I can roughly infer it, it would be better to explain it in the paper.
>
> Thanks for catching this. By total count, we meant the total number of intersection approaches (Figure 5). We added a more descriptive caption to the figure describing it.
>
>
> - For the benefit percentage, does 0.5 means 0.5% or 50%. If it is 0.5%, it seems the emission doesn't change a lot by switching to MARL control, and the key is that MARL agents can not achieve the same level of throughput as human drivers. However, it seems GCRL agent is close to human drivers' performance. I suggest giving more analysis and explanation to these figures. It can be done in the captions since I found the captions of Figure 5 and Figure 6 are simple copy-paste.
>
>
> We thank the reviewer for the suggestion. By "0.5," we meant 0.5%. As noted in the general response and in the footnote on page 9, we typically do not expect very high effective emission benefit percentages based on eco-driving literature. However, when applied at a large scale, even small reductions can lead to significant CO2 savings. These percentages also depend on the number of eco-driving agents, with literature suggesting that benefits increase with more agents. In this work, our focus is on CRL generalization, and we prioritize solving all context-MDPs in the CMDP for a given number of eco-driving agents, rather than emphasizing the absolute emission benefits.
>
>
> The reviewer is right in interpreting the failure cases arise from partly due to the MARL agents not being able to achieve the same throughput. Another reason is while throughput constraint is not violated, the emission is could be high. As we explained in the general response, we see that most MARL methods achieve positive effective emission benefits and sometimes outperforming GLOSA but there is still room for improvements as not all context-MDPs contribute to the overall benefits.
>
>
> - Also, is it possible to provide more visualizations in the appendix or even provide a video? So readers can know how different the map topologies and vehicle distributions are for these cities. I can not imagine a large dataset from pure text.
>
>
> We thank the reviewer for pointing this out. To address this, we provide a full comparison of how each city CMDP differs in terms of road topology and other factors, such as vehicle inflow distributions, in Figure 6, which compares the context feature distributions of incoming approaches across all ten cities. Additional details are provided in Appendix Section A.1. We also include representative visuals of reconstructed intersection scenarios in Figure 7, as well as the spatial distribution of intersections across cities in Figure 8. Additionally, in Appendix Section A.3, we analyze the suite of emission models by plotting emission landscapes to help readers understand the vehicular behaviors that could lead to emission benefits under the IntersectionZoo design. We hope these additions clarify the dataset and simulations for the reviewer.

---

> ### Author Response · Authors · 2024-11-22
> **Rebuttal by Authors (2/3)**
>
> - More experiments or results are needed. Though the experiments show that MARL can not generalize well, we don't know if it is due to the algorithm itself or if the task is too difficult to learn for RL agents, i.e. improper reward function and observation or MDP definition. One way to clarify this is to provide training performance as well. It needs to be proved first that the MARL can overfit the training scenarios and exceed human driver performance. Then we can confirm that the simulation design is appropriate, so MARL agents have the chance to surpass humans, and the failure on the test set is indeed due to poor generalizability.
>
> This is an excellent suggestion!! This comment from the reviewer prompted interesting discussion among the authors, which made us rethink the way we should present our experiment results. We believe this made the paper narrative stronger and for that, we are grateful.
>
> In the general response section titled “More analysis on benchmarking”, we revise the goals of our experiment to accommodate the reviewer’s suggestion and address the point the reviewer has made here. We request the reviewer to refer to that section for the details. In summary, following what the reviewer suggested, through our in-distribution generalization experiment in Section 6, we aim to answer two questions: 1) is the IntersectionZoo environment design (reward and observation functions) reasonable and 2) is there room for improvement for the MARL methods? Results show that most MARL methods can solve some parts of Atlanta and Salt Lake City CMDPs, indicating that most MARL methods learn to solve the problem to a certain extent, and the environment design thus appears to be reasonable. Further, given not all context-MDPs in each CMDP are solved, it shows there is still room for future algorithm improvement for better generalization.
>
> To further clarify our design choices, we added a new section in Appendix A.5 outlining our thought process behind the reward and observation functions, which we hope will help readers.
>
> We thought a lot about the reviewer's comment on the reward function design, and while we believe it is sufficiently justified by the supportive evidence from experiments and from past literature, we wanted to run a large reward coefficient search (to determine the scale of each reward term) beyond what we have considered so far to assess all possibilities. Given the significant computing needed for such a search and the short rebuttal period, we will update the paper (post rebuttal) in the subsequent iterations based on the results of the search.
>
>
> - Did the authors conduct experiments to see which factor affects the performance most?
>
> To answer reviewers' question about impactful factors, we conducted a Pearson correlation analysis of emission benefits with a few observed context factors that are known to impact emission benefits. The results are as follows.
>
> | Method | Lane length | Speed limit | Vehicle inflow | Road grade | Lane count |
> | ------ | ----------- | ----------- | -------------- | ---------- | ---------- |
> | GLOSA  | 0.098       | 0.08        | 0.079          | 0.06       | 0.019      |
> | PPO    | 0.082       | 0.041       | 0.037          | 0.036      | 0.015      |
> | DDPG   | 0.097       | 0.058       | 0.038          | 0.021      | -0.02      |
> | MAPPO  | 0.058       | 0.055       | 0.037          | 0.036      | 0.018      |
> | MADDPG | 0.082       | 0.034       | 0.034          | 0.018      | 0.004      |
> | GCRL   | 0.072       | 0.061       | 0.036          | 0.034      | 0.017      |
>
> We find that the better performing algorithms such as GLOSA, DDPG and MADDPG are more successful in intersections where there is more lane length. While other algorithms focus on intersections where there are higher lane lengths and speed limits. This indicates presence of different learning trajectories of different learning algorithms. We think the ability to debug generalization this way is a unique feature that IntersectionZoo provides.

---

> > ### Author Response · Authors · 2024-11-22
> > **Rebuttal by Authors (3/3)**
> >
> > - Also, a baseline can be added to use the IDM vehicle control these CVs to see if the MARL can outperform this simple rule-based baseline at least at training time. With such a trend, we can conclude it shows some signs that MARL can achieve better performance, and thus work on this benchmark is promising.
> >
> > We thank the reviewer for the suggestion to add a rule-based baselines. We note however that IDM is already used as a baseline for human-like driving in the paper. All emission benefit percentages are given as an improvement from IDM.  Therefore, we introduced a commercially implemented eco-driving controller GLOSA (Green Light Optimized Speed Advisory) [1] as a rule-based controller and MADDPG as another MARL method. GLOSA is a commercially implemented controller in modern vehicles, such as some models in Audi [2], and is considered a classical eco-driving method. Thus, we now have two rule-based baselines: a calibrated IDM model based on real-world data for human-like driving and a GLOSA controller as a rule-based eco-driving controller.
> >
> > In term of results, we see methods like DDPG outperform GLOSA controller in both Atlanta and Salt Lake City CMDPs. This confirms the reviewer’s point that with a trend that shows rule-based controller performance can be outperformed by some MARL methods, we can conclude that it shows some signs that MARL can achieve better performance, and thus work on this benchmark is promising.
> >
> > [1] Katsaros et al. Performance study of a green light optimized speed advisory (glosa) application using an integrated cooperative its simulation platform.
> >
> > [2] Audi expands traffic light information – now includes speed recommendations to minimize stops. URL https://media.audiusa.com/releases/301.
> >
> > - How would the temperature and humidity affect the policy behavior? For example, is it better to drive at low speed in summer to reduce the CO2 emission? But it would reduce the throughput right?
> >
> > The relationship between emission and temperature and humidity is a complex relationship that also depends on many other factors including vehicle type, age, fuel type, road grade etc. This can be seen in the emission model behavior analysis in Appendix Section A.3. Given this, the exact behavior of the vehicles are hard to predict. For the example the reviewer provided, it can be the case for some vehicle models under certain conditions but we can also see that it is not the case for some vehicles. For example, gasoline-driven passenger cars built in 2009 when travelling on a road with a grade of 0 degrees (Figure 9) tend to have lower emissions at higher speeds if the vehicle acceleration is small. In IntersectionZoo, what we expect is the benefits in expectation over all the these cases.
> >
> > - As far as I know, the OSM data only provides the location of the traffic light and doesn't tell which lane it controls. How does this information be completed in IntersectionZoo?
> >
> > The reviewer is right that OSM only provides the location of the traffic lights. Once we identified the location of the traffic signal and given we also know the lanes and their turn lane configurations, we set the traffic signal program based on sumo default and manually edit the programs if needed based on the programs used in similar intersections according to Automated Traffic Signal Performance Measures by Utah Department of Transportation. Once the phase set are found, we run an exhaustive search to find the optimal phase times.
> >
> > Lastly, we are very thankful to the reviewer for their constructive review. We hope our rebuttal addresses the reviewers' questions and suggestions. We would like to request the reviewer to consider increasing the score if our additional experiments and analysis are further satisfying to the reviewer. Thanks a lot for taking the time to review our work.

---

> ### Comment · Reviewer_dDfH · 2024-11-24
> **Thank you for the response**
>
> Thank you for the detailed reply. My main concern has been addressed, and the quality of this submission has been greatly improved. I believe it can benefit both the application and research of MARL x Transportation. I am increasing my score to 6.

---

> > ### Author Response · Authors · 2024-11-24
> > **Thank you!**
> >
> > Thank you for taking the time to go through our rebuttal. We are glad that the reviewer found it to address their concerns. Thanks to the reviewer's constructive review, the submission has indeed improved greatly, and for that, we are grateful. Thanks again for increasing our score!!

---

### Author Response · Authors · 2024-11-22
**Rebuttal by Authors**

# General Response (3/3)

## More details of the context-MDPs and traffic scenarios

We thank the reviewers for pointing out where we can provide more details on the context-MDP definition and traffic scenarios. Following these suggestions, we added a separate sections in Appendix Section A.5 to describe our rationale behind the reward function and observation function of each context-MDP and for more details on traffic scenarios including comparison of different factor distributions in each CMDP (Appendix Section A.1 and Figure 6), the emission model behaviors (Appendix Section A.3 and Figure 9) and some illustrations of the intersection topology diversity (Figure 7) and the spatial distribution of the intersections considered (Figure 8). Further we added a section titled “Guidelines for using IntersectionZoo” with some general guidelines for users to make the use of IntersectionZoo more standardized.

[1] Katsaros et al. Performance study of a green light optimized speed advisory (glosa) application using an integrated cooperative its simulation platform.

[2] Audi expands traffic light information – now includes speed recommendations to minimize stops. URL https://media.audiusa.com/releases/301.

[3] Mintsis et al. Dynamic eco-driving near signalized intersections: Systematic review and future research directions.

---

### Author Response · Authors · 2024-11-22
**Rebuttal by Authors**

# General Response (2/3)

### Evaluations and results

We recognized that our initial evaluation metrics and result format may have caused confusion. To clarify and streamline evaluations, we introduced a composite metric, effective emission benefits, alongside the individual metrics of emission and throughput benefits. This metric provides a single score for each method across the context-MDPs in a CMDP, reflecting emission improvements relative to human-like driving (IDM) while considering only approaches where emissions are reduced without sacrificing throughput. If either throughput or emissions worsen compared to IDM, we revert to human-like driving, resulting in zero benefits. The formula for this metric is outlined in Section 5.3. We used this metric to evaluate the GLOSA controller and MARL methods, with results provided below, all relative to the IDM baseline.

| Method           | Atlanta's effective emission benefits ($\uparrow$) | Salt Lake City’s effective emission benefits ($\uparrow$) |
| ---------------- | ---------------------------------------------------- | ----------------------------------------------------------- |
| IDM (baseline)   | 0.0%                                                 | 0.0%                                                        |
| GLOSA (baseline) | 2.30%                                                | 0.95%                                                       |
| PPO              | 0.47%                                                | 0.05%                                                       |
| DDPG             | 2.84%                                                | 2.95%                                                       |
| MAPPO            | 1.11%                                                | 0.61%                                                       |
| GCRL             | 0.0%                                                 | 0.0%                                                        |
| MADDPG           | 4.47%                                                | 0.0%                                                        |


As shown in the table, most MARL methods achieve positive benefits, indicating that some context-MDPs in the CMDP have been solved. Methods like DDPG and MADDPG even outperform GLOSA in the Atlanta CMDP, though MADDPG struggles in the Salt Lake City CMDP. We hope this revised way of presenting the benchmark results clarifies that the most MARL methods are indeed learning something meaningful, while at the same time indicating that there is room to improve.

We would like to clarify the results presented in Figure 5 (Atlanta CMDP) and Figure 10 (Salt Lake City CMDP), as there appears to be some confusion about their interpretation. These figures show the effective emission benefit histograms for each city’s intersection approaches, where the throughput reduction constraint is satisfied. Specifically, if the throughput under the learned policy is worse than the human-driving baseline, we set the emission benefit for that approach to zero (defaulting to human-like driving). If throughput is not reduced, we retain the emission benefits, whether positive or negative. With this interpretation, it's important to note that most methods solve some context-MDPs (i.e., >0 emission benefits) in each CMDP, so the plots should not be seen as a complete failure of the methods. We have updated Section 6 to reflect this clarification.

---

### Author Response · Authors · 2024-11-22
**Rebuttal by Authors**

# General Response (1/3)

We thank the reviewers for their insightful and thoughtful feedback. We are encouraged that you found our work to have a significant impact on advancing the research on contextual reinforcement learning **(Ld56, r4Ef, giey)**, multi-agent reinforcement learning **(dDfH, Ld56)**, and has a real-world impact **(dDfH, Ld56, r4Ef, giey)**. Reviewers found the IntersectionZoo to be a well-built environment for MARL researchers to make a real-world impact **(dDfH, Ld56, giey)**, and the transportation community could also benefit from it **(Ld56)**. We are glad that all reviewers unanimously found that the data-driven simulation of real-world traffic scenarios is well-detailed and sufficiently capture real-world complexities and variabilities to study generalization in RL **(dDfH, Ld56, r4Ef, giey)**. The comprehensive nature of IntersectionZoo, which contains a dataset, a simulator, evaluation metrics, and a training framework, is appreciated by the reviewers **(dDfH, giey)**. Reviewers also found the scale of the contextual MDPs we provide impactful **(dDfH, Ld56, giey)** and noted the paper is well-written and straightforward to follow **(Ld56, r4Ef)** and that the code is well-documented **(dDfH)**. We are glad that reviewers found our comparison of IntersectionZoo with the existing benchmarks thorough and well-summarized **(Ld56, r4Ef)**. Reviewers also agree that the experiments conducted with the popular MARL algorithms strengthen the case for IntersectionZoo **(dDfH, r4Ef)**.

On the constructive side, reviewers mentioned that more analysis of the benchmarking results could strengthen the work better, particularly focusing on adding a rule-based baseline and more MARL algorithms or analysis of the existing algorithms and clarifying the rationale behind the benchmarking section.

## More analysis on benchmarking

We thank the reviewers for their valuable feedback on the benchmarking analysis. This response summarizes our overall revisions. The paper has been updated accordingly.

### Goals of experiments

We thank the reviewers for their insightful comments on the experimental section. Based on their feedback, we clarified the goals of our experiments and added further analysis to strengthen the section.

As reviewers **r4Ef** and **grey** suggested to clarify, we would like to clarify that our goals in evaluating the MARL methods are threefold. First, we aim to show that popular RL algorithms used in multi-agent control demonstrate signs that they do not appear to easily solve the CRL problem presented in IntersectionZoo CMDPs. This highlights that CMDPs presented in IntersectionZoo provide room for reasonable algorithm improvements. Second, we show that some MARL algorithms solve the presented CMDPs to some extent, though not completely (this means some context-MDPs in a CMDP have positive emission benefits and others do not). This highlights that the IntersectionZoo CMDPs are designed reasonably well (e.g., reward function, observation function, etc.) and MARL methods can learn and solve the problem to a certain extent. Finally, we also aim to show examples of cases in which IntersectionZoo facilitates different experiments on targeted evaluations of generalization — an aspect that not many benchmarks have facilitated. For this, we assess the systematicity and productivity in generalization as given in Appendix Section B.4. We note, as some reviewers acknowledge, our goals in experiments are not to extensively benchmark MARL methods but to achieve the three goals stated above.

### Benchmarking analysis with more methods
In response to reviewer suggestions, we added a rule-based baseline and more MARL baselines. Specifically, we introduced GLOSA (Green Light Optimized Speed Advisory) [1], a commercially implemented eco-driving controller used in vehicles like Audi [2], as a rule-based baseline, alongside the calibrated IDM model. We also added MADDPG as an additional MARL method. This gives us two rule-based baselines (IDM and GLOSA), as now described in Section 5.2, and five MARL algorithms for comparison.

---

### Meta-Review · Area_Chair_zHU6 · 2024-12-27

**Metareview:**

The paper presents a benchmark for MARL via fleet (CO2) control.

The primary strength of this paper is the extent to which it builds on real-world data and captures important features of real world deployment.

The reviewers noted several weaknesses. First, the reviewers rightly question whether this benchmark could be extended to a broader class of problems. There were also several questions around poor/lacking analysis of existing methods, but that has largely been addressed to the satisfaction of the reviewers.

Reviewers all recommend acceptance, and the major weaknesses have been discussed and largely addressed by the authors. Benchmarking work is often severely underrated by the community and this work involves significant data collection and simulation preparation, and so I lean toward acceptance.

**Additional Comments On Reviewer Discussion:**

There was a fairly substantial discussion around the extent to which the benchmark could be broadened. The authors noted the fragility of MARL methods as one reason why it is difficult to arbitrarily expand the benchmark. This is reasonable, but the authors should look into this direction further.

Several reviewers discussed the quality of presentation, and this has largely been fixed.

Several reviewers asked for more benchmarking of existing methods. The authors have substantially broadened this section.

---

### Decision · Program_Chairs · 2025-01-22

Accept (Poster)